# Discovering the genes mediating the interactions between chronic respiratory diseases in the human interactome

Enrico Maiorino [1,2]*, Seung Han Baek[1], Feng Guo [1], Xiaobo Zhou [1], Parul H. Kothari[1], Edwin K. Silverman[1], Albert-László Barabási [1,2], Scott T. Weiss[1], Benjamin A. Raby[1] & Amitabh Sharma [1,3]

The molecular and clinical features of a complex disease can be influenced by other diseases affecting the same individual. Understanding disease-disease interactions is therefore crucial for revealing shared molecular mechanisms among diseases and designing effective treatments. Here we introduce Flow Centrality (FC), a network-based approach to identify the genes mediating the interaction between two diseases in a protein-protein interaction network. We focus on asthma and COPD, two chronic respiratory diseases that have been long hypothesized to share common genetic determinants and mechanisms. We show that FC highlights potential mediator genes between the two diseases, and observe similar outcomes when applying FC to 66 additional pairs of related diseases. Further, we perform in vitro perturbation experiments on a widely replicated asthma gene, *GSDMB*, showing that FC identifies candidate mediators of the interactions between *GSDMB* and COPD-associated genes. Our results indicate that FC predicts promising gene candidates for further study of disease-disease interactions.

[1] Channing Division of Network Medicine, Brigham and Women's Hospital, Harvard Medical School, Boston, MA, USA. [2] Network Science Institute, Center for Complex Network Research, Department of Physics, Northeastern University, Boston, MA, USA. [3] Deceased: Amitabh Sharma. *email: emaiorino@bwh.harvard.edu

Biological networks are powerful resources for discovering and understanding the mechanisms that underlie human complex diseases[1,2]. Indeed, it is accepted that biological components such as genes and proteins do not act in isolation, but are connected through intricate networks of molecular interactions that allow perturbations to diffuse across the system and generate, enhance or alter the disease phenotype. Over the last decade it has been observed that protein-coding genes associated to a disease have a strong tendency to interact with each other and agglomerate in a specific network neighborhood called the disease module[3–6]. However, disease progression is strongly influenced by the biological context of the organism. Perturbations causing one disease might affect other diseases, especially when the involved genes lie in the same network neighborhood, producing complex phenotypes and comorbidities[7].

Finding the molecular commonalities between related diseases is crucial in understanding their heterogeneity as well as identifying common biomarkers and therapeutics. As a step in this direction, Menche et al.[5] measured the network-based separation between 226 disease pairs, observing that overlapping disease modules display significant molecular similarity, elevated coexpression of their associated genes, similar symptoms and high comorbidity. However, while the introduced separation measure offers information on the similarity of two diseases, it does not help in identifying the genes encoding proteins that influence both diseases. Furthermore, mediator genes may not be part of either disease module, but they could mediate the interactions between the two diseases without participating in the core pathways of the individual diseases. In this work we propose a methodology to identify the mediators linking pairs of complex diseases, focusing on asthma and chronic obstructive pulmonary disease (COPD), two of the most widespread chronic respiratory diseases that have been estimated to be the cause of over 3 million deaths worldwide[8]. Asthma and COPD are influenced by genetic and environmental factors and they often manifest through similar phenotypes, like airflow obstruction, inflammation, and shortness of breath[9,10]. A widely-accepted definition of their differences is still lacking since many cases fall in-between the two classic descriptions of these conditions, and patients often show asthma-like and COPD-like features simultaneously. For example, airflow obstruction reversibility, considered one of the main hallmarks of asthma, can be present in many COPD patients[9,10]. On the other hand, fixed airflow obstruction, a cardinal manifestation of COPD, can develop in asthmatics as well, particularly those with severe disease or persistent symptoms since childhood[11,12]. Moreover, people affected by asthma since birth are more likely to develop COPD at later ages[13–15]. This phenotypic gray area has been the source of extensive debate on a possible common genetic origin of the two diseases, a hypothesis first proposed by Orie and Sluiter[16], and termed the "Dutch hypothesis". Despite the considerable effort in delineating and summarizing the richness of the clinical manifestations of asthma and COPD, there is still little understanding of the shared molecular mechanisms and the causal relationships between the two disorders. Next-generation sequencing and genome-wide association studies (GWAS) allow to identify potential causal genes that can explain the development of these chronic respiratory diseases and possibly offer mechanistic insights into their shared causality[17,18]. Although the presence of shared disease gene associations might be expected in the context of the asthma-COPD overlap, previous work has provided little genetic support for the Dutch hypothesis, finding little to no overlap between the major asthma and COPD genes identified via GWAS[12]. Here we show that network-based statistical methods can provide additional avenues to explore this problem.

We model asthma and COPD in the network of protein–protein interaction (PPI), also referred to as the interactome. Each node of the network corresponds to a protein-coding gene and the link between two genes represents a physical interaction between the corresponding proteins. In order to find the mediators between the two diseases we define a topological measure, called flow centrality (FC), identifying the genes that are involved in most of the molecular interactions occurring between the two disorders. We show that flow central genes are more functionally related with each other and with the disease genes of asthma and COPD than expected by chance. Furthermore, we generalize these results by replicating it on 66 additional pairs of related diseases. Using multiple lines of evidence, including prior literature, gene coexpression analysis in multiple transcriptomics datasets from asthmatic and COPD subjects, and in vitro genetic perturbation in a bronchial epithelial cell line (a cell type relevant to both asthma and COPD), we show that genes with high FC values are biologically meaningful and related to known asthma-specific, COPD-specific and overlapping processes. Together, these results establish flow centrality as a valuable tool in the detection of genes mediating the interaction between different diseases, offering an opportunity to understand the relation between complex diseases.

## Results

**Disease modules construction.** We considered the protein–protein interaction constructed previously[19], which integrates high-quality yeast-two-hybrid data from publicly available datasets and literature-derived interactions (see Methods). While a gene may express different isoforms, we only considered one protein product per gene, and thus we refer to the nodes of the network as genes or proteins interchangeably throughout the text. We compiled two sets of seed genes representing known GWAS loci associated to asthma and COPD from the recent literature (see Methods). The asthma seed gene set is composed of 36 genes (35 mapped in the network) and the COPD gene set is composed of 30 genes (Supplementary Data 1 and 2, respectively), and the two sets have no overlap. To explore the network neighborhood of each disease we construct a disease module by applying the DIAMOnD algorithm, a procedure for ranking the genes in the network according to their connectivity significance to the seed genes[20] (see Methods). To define a cutoff for the gene ranking calculated through DIAMOnD, we considered two reference sets of GWAS-significant genes associated, respectively, to asthma and COPD, downloaded from the UK-Biobank repository[21] (UKB). For both the diseases, the final module size was chosen as the size that maximized the enrichment of UKB genes in each respective module (see Methods). The two modules have 14 overlapping genes (see Supplementary Fig. 1b), summarized in Supplementary Data 3. Most of the overlapping genes in the list, such as *TP53, MDM2, NFKB1, RELA, CTNNB1, TGFBR2, SMAD3, MAPK1, MAPK3, MAPK8, STAT1,* and *STAT3* are known to be involved in the regulation of apoptosis, proliferation, inflammation, cellular remodeling and differentiation[22–26]. Although these biological processes may play a role in asthma and COPD, they are not unique to these disorders. This inherent non-specificity can also be deduced by the high degree that characterizes all these genes, as shown in Supplementary Data 3. Furthermore, the empirical $p$-value quantifying the significance of their overlap is largely non-significant ($\sim 0.39$), confirming the elusive nature of the asthma-COPD relationship. This lack of significance in the overlap motivated our following analysis.

**Flow centrality between modules.** Asthma and COPD manifest through similar phenotypes and symptoms, and many asthma patients develop COPD at older ages[9,10,12]. This observation suggests that a perturbation originating from asthma-specific genetic risk factors may slowly disrupt critical pathways, ultimately leading

to the development of COPD in susceptible subjects. This perturbation may not be carried exclusively by the direct interactions of disease-specific genes; it may in fact travel through mediating genes that are not specifically linked to a single disease, thus making their recognition with standard approaches challenging.

These mediating genes are likely to be participating in the majority of the interactions between the two modules, constituting a "bottleneck" in the communication between the two diseases. In a network, betweenness centrality measures quantify the frequency of occurrence of a node in the paths that connect all the other nodes. A path is defined as an ordered sequence of steps across the edges of the network that start from a source node and lead to a destination node. There are multiple possible paths between any source and destination, and several works in literature have been dedicated to exploring different criteria for selecting and weighting these paths. For example, the classic betweenness centrality measure, proposed by Freeman[27], considers only the shortest paths between the source and destination nodes. In other work a random walk betweenness centrality is proposed, where paths are weighted by the probability of being traversed by a walker in a random walk process[28]. Further, in another study, the authors designed a factorial weighting scheme that favors paths of shorter lengths, called communicability betweenness[29]. Kivimaki et al.[30] defined the framework of randomized shortest paths (RSP), which interpolates between the classic concept of shortest-path-based betweenness centrality and the random walk betweenness centrality through a temperature parameter. The canonical form of these measures is an average across all the paths starting from any source node and leading to any destination node, resulting in an estimate of the node's centrality in the global network topology. While betweenness-central nodes may have a role in the pathways of asthma and COPD, by definition they are not specific to these two disorders (since their centrality does not change when considering different diseases), and thus they are less likely to provide meaningful information about their shared pathways.

In this work we introduce the concept of flow centrality, explained in detail in the Methods section (see Fig. 1a). Flow centrality is a betweenness measure that is parametric on a source set and destination set of nodes, and its coverage spans exclusively the shortest paths connecting the two modules, instead of the whole network, similarly to a recently proposed measure called Double Specific Betweenness (S2B)[31]. Therefore, when all the nodes of the network are selected as both sources and targets of the shortest paths flow centrality reduces to the classic betweenness centrality defined in ref. [27]. Flow centrality and the betweenness centrality measures described above are correlated to the node degree, regardless of the chosen source and target modules. To correct for this effect we defined a randomization scheme of the source and target modules to generate a null distribution of expected flow centrality values. The flow centrality score (FCS) is then calculated as the z-score of the flow centrality value when compared with the null distribution (see Fig. 1b and Methods section). A large positive value of the FCS implies that the node is highly central with respect to the source and target gene sets, even when accounting for its global centrality.

By defining the asthma node set as the source module and the COPD node set as the destination module, we calculated the flow centrality scores of all the nodes of the network. While all the betweenness centrality measures are highly correlated to the degree and with each other (Spearman's $\rho = 0.91 \pm 0.07$, see Supplementary Figs. 2 and 3), denoting low specificity with respect to the asthma and COPD modules, we find that the flow centrality scores are quite orthogonal to the other measures (Spearman's $\rho = -0.22 \pm 0.04$), suggesting that FC is highly specific of the particular source and target gene sets.

Among the top flow central nodes (see Supplementary Data 4), several genes, such as *SLC39A8*, *SOX17*, and *MFAP4* show a direct relationship with asthma and COPD. More specifically, it has been found in literature that the expression levels of *SLC39A8*, *SOX17*, and *MFAP4* might directly affect both asthma and COPD. For example, *MFAP4*-deficient mice showed attenuated eosinophilic inflammation, eotaxin production, airway remodeling and airway hyper-responsiveness that are classical characteristics of asthma, while expression of *SOX17* in respiratory epithelial cells decreased the expression of transforming growth factor-beta (*TGF-β*)-responsive cell cycle inhibitors such as p15, p21, and p57 in the adult mouse lung[32,33]. *SOX17* also inhibited *TGF-β*-mediated transcriptional responses in vitro, demonstrating an inhibitory effect on the *TGF-β* pathway[32,33]. *TGF-β*, that is highly expressed in small airway epithelium of COPD patients[34], is known to play a role in the increased submucosal collagen expression occurring within the disease, and is also known to be a mediator involved in tissue remodeling in the asthmatic lung[35,36]. *SLC39A8*, a zinc transporter, is a major portal for cadmium (Cd) uptake[37]. *SLC39A8* mRNA and protein expression levels were found to be significantly increased in lungs of chronic smokers compared with nonsmokers[37]. Cd is found in cigarette smoke, and it could contribute to smoking-induced lung diseases such as COPD[37]. In the presence of Cd, inhibition of the *NF-κB* pathway and *SLC39A8* expression reduces cell toxicity while *TNF-α* treatment of primary human lung epithelia and A549 (lung cancer cell line) cells showed induced expression of *SLC39A8*, resulting in higher cell death[37,38]. *IHH* and *DHH* are part of the sonic hedgehog pathway and are known to directly interact with *HHIP* (hedgehog interacting protein) which is strongly associated with the risk of COPD[39,40]. *HHIP* competes with *Ptch1* (which is the membrane receptor for *IHH*) for the binding of *IHH* and *DHH*. *Ptch1* binding to *IHH* and *DHH* triggers the hedgehog signaling pathway, therefore the binding of *HHIP* with *IHH* negatively regulates the hedgehog pathway which is known to have a crucial role in lung development[39,41].

**Functional similarity of flow central genes**. To validate the biological relevance of flow central genes, we selected the shortest paths between asthma and COPD seed genes whose intermediate nodes (i.e., all the nodes in the path except for the source and target) are characterized by high FCS (see Methods section for further details on the selection). By applying this selection criterion we obtained 371 distinct central paths to which we refer to as flow central paths (see Fig. 1c).

We assessed the degree of functional relatedness between the genes occurring in the flow central paths by considering their associated Gene Ontology (GO) terms. The GO similarity between two genes is defined as the best-match average (BMA) of Resnik's similarity measure, one of the most well-known information-based similarity measures for hierarchically-ordered elements[42]. Further, we defined the sequential similarity (SS), a path-level quantity that measures the average GO similarity between adjacent genes in a network path (see Fig. 1d top left and Methods section). The higher the SS, the more functionally similar are the genes along the path.

We calculated the SS for each flow central path, obtaining a distribution of 371 similarity values. To estimate their significance we generated two null distributions of network paths, namely the random paths of Type A and Type B. To generate the Type A set we extract 10,000 random paths with a distribution of lengths that matches the empirical distribution observed in the FC paths (length-preserved) using the randomization scheme explained in Methods. The Type B set is constructed by randomly extracting

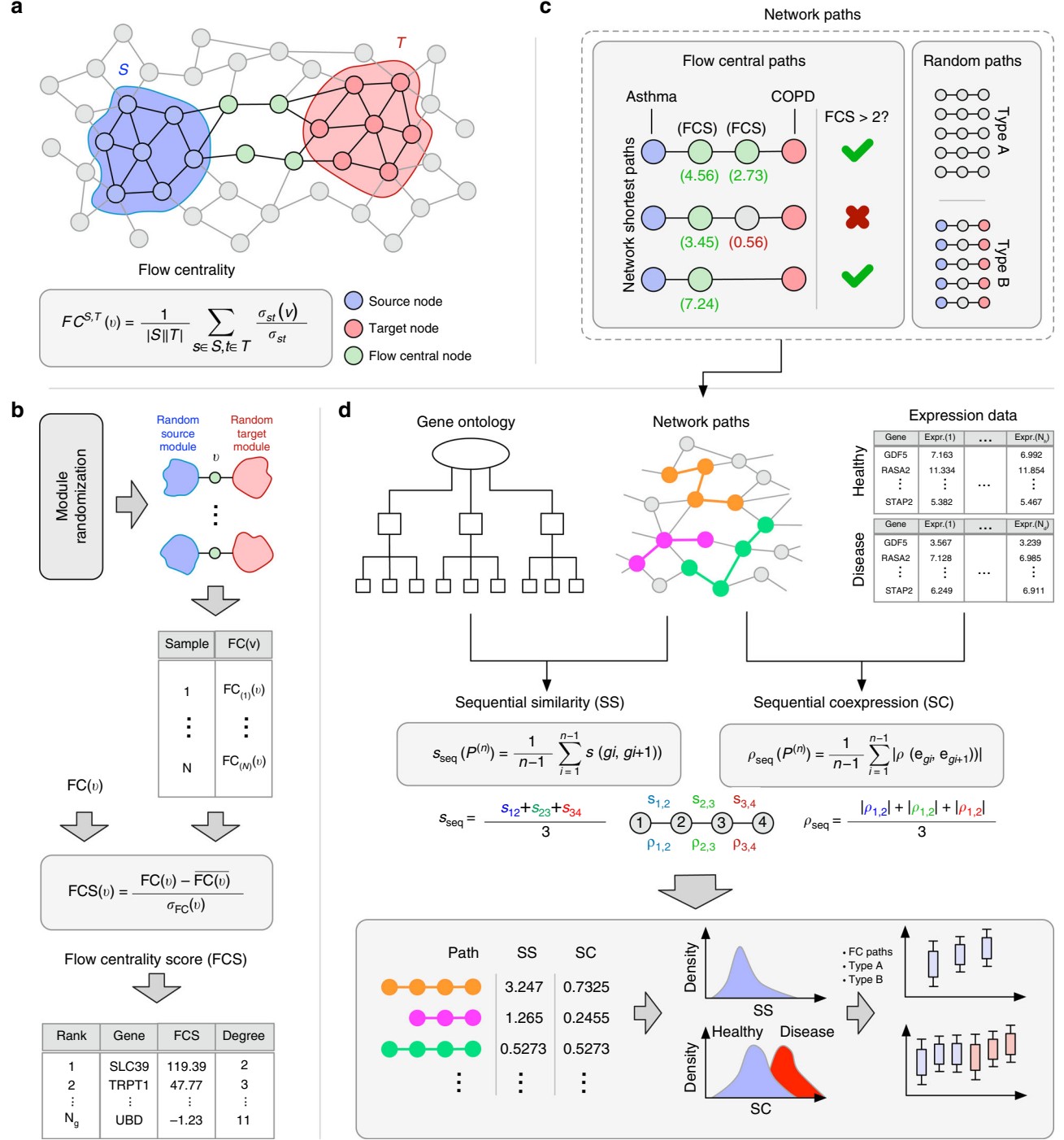

**Fig. 1 Overall scheme of the analysis. a** Flow centrality measure. The source nodes (blue) are connected to the target nodes (red) preferentially through high flow centrality nodes (green). **b** Flow centrality score calculation. 1000 samples are generated through the randomization scheme explained in Methods section. For each node, its FC score is compared with its corresponding values in the random samples and a $z$-score is obtained, defined as flow centrality score (FCS). **c** Flow central paths are selected among the shortest paths between asthma and COPD seed genes. The selection condition is that all the intermediate genes in the paths have a FCS >2. Two sets of 10,000 random paths are extracted (Type A, length-preserved, and Type B, endpoints-preserved), for a total of three sets of paths. **d** starting from each sets of network paths, GO annotations and GEO expression data, the sequential similarity and sequential coexpression values are calculated and compared.

10,000 paths from the pool of the shortest paths between asthma and COPD seed genes (endpoints-preserved). Type A accounts for the possible biases related to the particular lengths of the FC paths, while Type B allows a direct comparison to the case where no FC information is utilized.

Figure 2a shows the comparison of the SS distributions for the flow central, Type A and Type B paths. The sequential similarities of FC paths are considerably greater than the similarities of Type A and Type B paths (one-tailed Mann–Whitney test $p$-values 1.12e−111 and 2.06e−77, respectively). We evaluated the separate

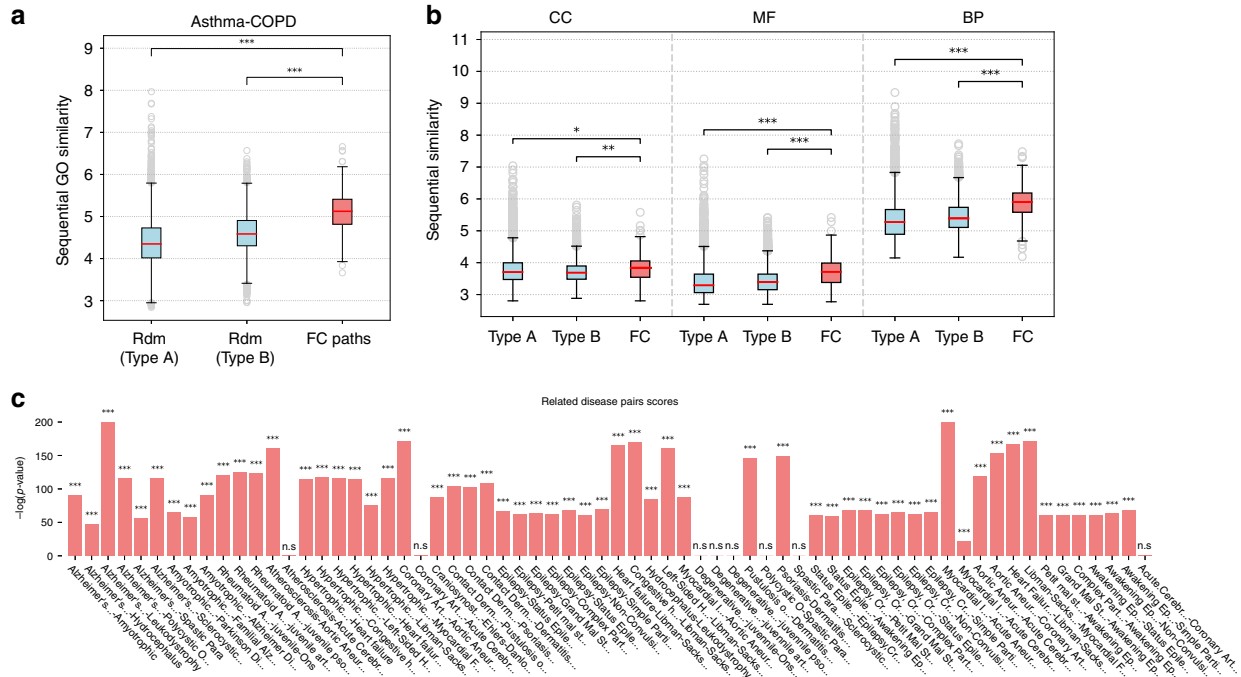

**Fig. 2 GO similarity of flow central paths. a** Distribution of sequential GO similarities (SS) of random paths of Type A, Type B, and FC paths; **b** SS of Type A, Type B, and FC paths calculated for the three main GO root terms: cellular component (CC), molecular function (MF), and biological process (BP). **c** Worst-case *p*-values of comparison between FC paths and random paths for each related disease pair. In the boxplots, boxes indicate the quartiles, whiskers extend to an additional 1.5 * IQR interval, and the medians are highlighted in red. One, two, and three asterisks, respectively, denote a Mann–Whitney *p*-value that is <0.05, 1e−4, and 1e−10, and "n.s." stands for a non-significant result.

contributions of the three main Gene Ontology categories to the global similarity (see Fig. 2b): cellular component (CC), molecular function (MF), and biological process (BP). In all cases the similarities of FC paths are significantly higher than expected. In Fig. 3a we show the FC paths ordered by number of GO annotations and the top 50 BP GO terms ordered by their information content, i.e., their specificity in the entire GO database. Biological regulation is one of the most enriched categories, which is expected because of the large number of genes annotated to regulatory processes. However, its occurrence is still more frequent than cellular process terms which are more common in the GO annotation corpus, suggesting the importance of regulatory mechanisms in the cross-talk between asthma and COPD pathways. For example, in Figs. 3b–d are shown three FC paths that are enriched in several biological processes which are relevant for both the disease onset and exacerbation. Regulation of chemokine production, regulation of T-cell activation, wound healing, tube development and inflammatory response are biological processes that are involved in airway remodeling and immune response for both asthma and COPD. More specifically, the genes of the paths in Fig. 3b, c are highly related to the *TGF-β* signaling pathway. The *TGF-β* signaling pathway, which consists of proteins such as *TGFBR1*, *TGFBR2*, *SMAD2*, and *SMAD3*, is involved in differentiation, cell growth and many other cellular functions that play a crucial role in development and wound healing[43,44]. The *RAR* pathway, which interacts with the *TGF-β* signaling pathway through the *SMAD* proteins, is activated by binding retinoic acid to the retinoic acid receptors (RARs) such as *RARB*[45,46]. The *RAR* pathway is also involved in cellular functions that play a crucial role in development and wound healing[45]. On the other hand, the FC path shown in Fig. 3d consists of genes that are involved in the inflammatory response through the *JAK-STAT* signaling pathway and the *TLR4* signaling pathway[47,48]. Both the *JAK-STAT* signaling

pathway and the *TLR4* signaling pathway play a crucial role in immune response and the cross-talk between the two pathways is thought to regulate the severity of the host inflammatory response[49].

**Functional similarity of FC genes of related diseases**. To test whether the previous result holds in general, we considered the corpus of gene-disease associations (GDA) contained in the DisGeNet repository[50] and the disease–disease similarities extracted from the Disease Ontology knowledge base. We selected all the pairs of similar diseases with a minimum of 50 associated genes and low overlap as to reduce to a case similar to asthma and COPD (see Methods section and Supplementary Figs. 6 and 7). These criteria result in 66 distinct pairs of diseases that are related according to their phenotypes, genetic causes, localization in the organism, etc (Supplementary Data 5). Some examples are Alzheimer's disease and amyotrophic lateral sclerosis, that are both neurodegenerative diseases which share similar phenotypical features such as dementia, language dysfunction, and muscle weakness[51,52], and pathologic processes involving genes playing a major role in protein homeostasis and endoplasmic reticulum stress[53,54]; psoriasis and allergic contact dermatitis are both inflammatory skin diseases involving the immune response that share similar phenotypical features due to inflammation[55,56] and pro-inflammatory pathways involving IL-36γ[57]; polycystic ovary syndrome and Alzheimer's disease do not share phenotypical features, yet studies showed that the two diseases might have a casual relation based on insulin resistance and through the protein phosphatase 2A pathway[58–60]. For each pair, we calculated the flow centrality of all the nodes in the network, selected their corresponding FC paths and extracted 10,000 Type A and B paths, following the same scheme defined above. We proceeded to

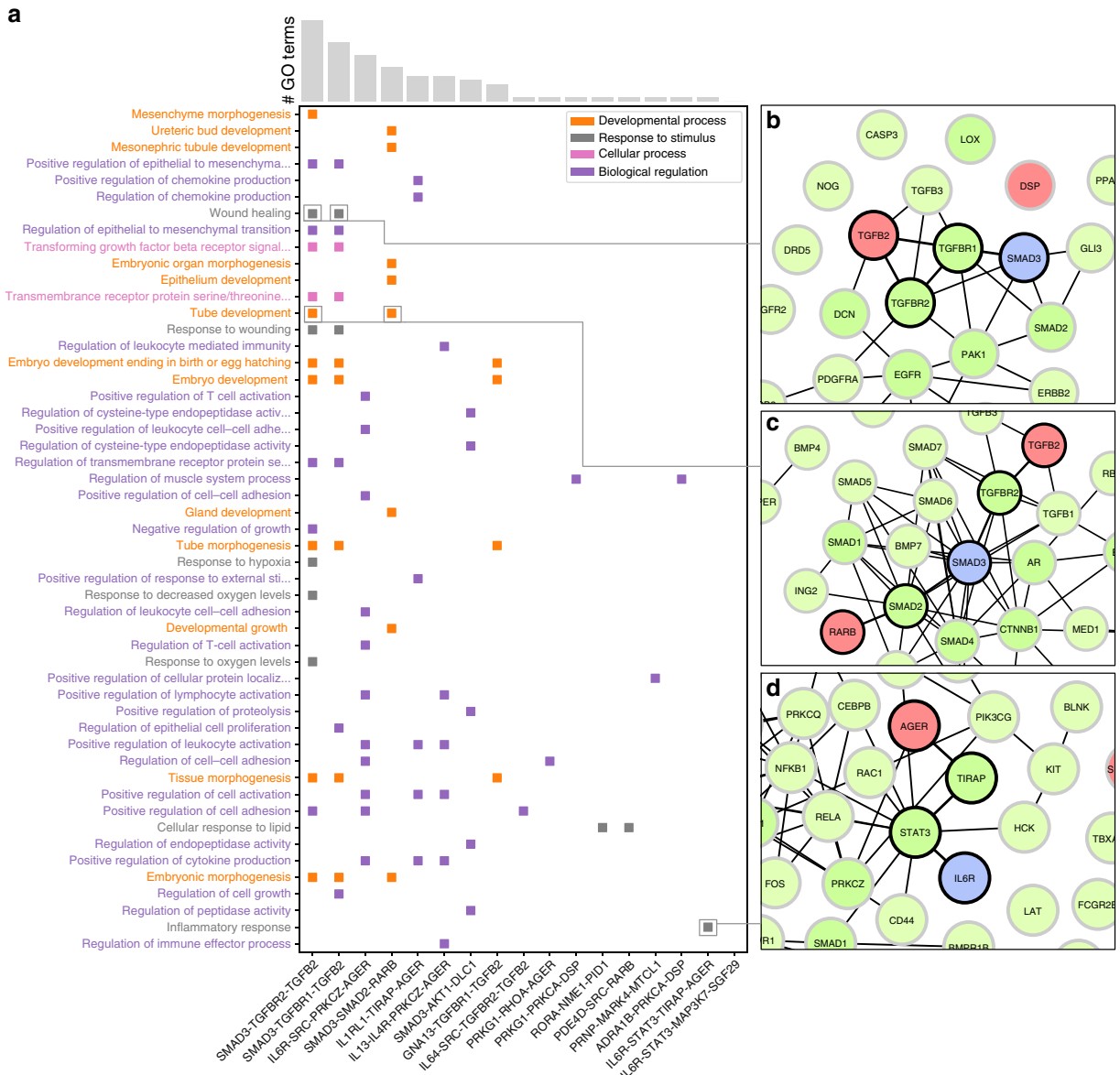

**Fig. 3 Top 50 biological process GO terms.** Top 50 biological process GO terms enriched in flow central paths, ordered by information content, and top 17 FC paths ordered by number of GO annotations. **a** Squares indicate that the GO term on the left is annotated to all the genes in the FC path on the bottom. Different colors indicate different subclasses of the biological process category; **b–d** examples of paths corresponding to selected GO terms.

evaluate the SS values of FC paths and Type A/B paths, computing two p-values $p_A$ and $p_B$, corresponding, respectively, to the comparisons FC ↔ Type A paths and FC ↔ Type B paths. Then, we classified every disease pair with its least significant p-value (i.e., $\max(p_A, p_B)$), determining a worst-case estimate of the SS increase in FC paths. The scores of the resulting p-values, computed as their negative log-transformed values, are shown in Fig. 2c. We find that for the vast majority of disease pairs (58 out of 66) we obtain highly significant differences (p-value < 1e−20) between the SS of FC paths and random. In addition, we tested the specificity of the previous result. We generated 100 random degree-preserved sets of nodes of each disease module occurring in the 66 pairs (6600 pairs of random modules). For each original disease pair, we compared its SS distribution to each random pair through Mann–Whitney test, obtaining 100 worst-case p-values (see Methods). We find that the FC paths of the original disease pairs are almost always more sequentially similar than their randomized counterparts (Supplementary Fig. 8), with the

only exception being the disease pair Hydrocephalus ↔ Leuko-dystrophy, possibly due to a weaker genetic link between the two diseases. Overall, this result shows that flow centrality is a highly specific property of the source and destination modules, and that it would not yield the same outcomes if applied to unrelated genes.

**Coexpression of flow central genes**. To highlight the putative mechanistic connections between asthma and COPD, we measured the coexpression of the genes along the flow central paths connecting the two diseases. Although gene coexpression does not necessarily imply a functional relation, it indicates whether two genes are synergistic (or antagonistic) in terms of expression, suggesting co-participation in the same biological processes. Thus, a higher degree of coordination between FC genes with asthma and COPD disease genes indicates their involvement in biological processes common to both diseases.

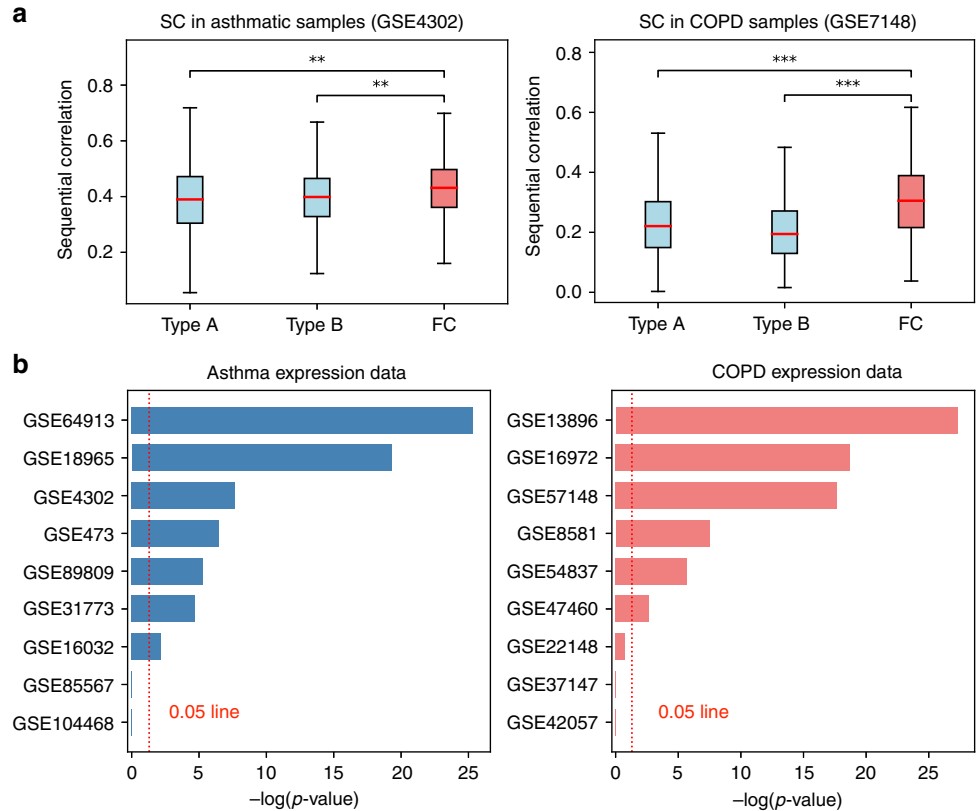

**Fig. 4 Sequential coexpression of flow central paths. a** Distributions of sequential coexpression values of the flow central (FC) paths compared with random paths of Type A and B for asthma (left) and COPD (right). **b** Worst-case *p*-value scores for in asthma (left) and COPD (right), calculated across all the disease classes in each GEO dataset. In the boxplots, boxes indicate the quartiles, whiskers extend to an additional 1.5 * IQR interval, and the medians are highlighted in red. One, two, and three asterisks, respectively, denote a Mann–Whitney *p*-value that is <0.05, 1e−4, and 1e−10, and "n.s." stands for a non-significant result.

As reference expression data we considered two expression datasets of asthmatic and COPD patients from Gene Expression Omnibus. The first dataset is a microarray expression measurement of airway epithelial cells in asthmatics and healthy controls (GSE4302[61]), and the second one is an RNA-seq profiling of lung tissue in COPD patients and healthy controls (GSE57148[62]) (see Supplementary Data 6 and Methods section for details). To measure the coexpression of the genes along each path, we defined the sequential coexpression (SC) as the average absolute coexpression between adjacent genes in the path (see Methods). For a given path, a higher sequential coexpression denotes a larger degree of coexpression between the genes interacting along the path. For each expression dataset, we calculated the SC of the FC paths for the healthy and disease states separately (Fig. 1d, e), obtaining two distributions of SC values for asthma and COPD, respectively. In the same way, we evaluated the SC values of the Type A and Type B paths for the same cases described above (asthma control/disease and COPD control/disease).

We find that in both the asthma and COPD data the FC paths are enriched for statistically higher SC values compared with both Type A paths (MW *p*-values 8.38e−10 and 2.14e−18, respectively) and to Type B paths (*p*-values 2.25e−8 and 1.41e−33, see Fig. 4a). In addition, the same result holds in the samples of healthy patients (worst-case *p*-value < 1e−9), suggesting that FC paths correspond to interaction cascades that can be active both in healthy and disease state.

We repeated the same analysis in 16 additional GEO expression datasets. In each dataset, several subdivisions of the disease and healthy samples (classes) were considered when further information was available (such as cell type, tissue, or

disease severity, see Supplementary Data 6). Similarly as before, we classified every dataset with its least significant *p*-value across all the classes. The SC values and the scores of the resulting *p*-values are shown, respectively, in Supplementary Fig. 9 and Fig. 4b. Despite the large variability of the considered expression datasets, we find similar outcomes for all the disease classes in a total of 13 out of 18 GEO datasets, with five cases being largely significant (worst-case *p*-value < 1e−10). These results suggest that the interaction paths identified by flow centrality are robust to fluctuations and are not specific to a single cell type, tissue or experimental setting. Interestingly, we observe that the same result holds also in the respective classes of the healthy or control states (see Supplementary Fig. 10).

Since asthma and COPD are related, we hypothesized that their flow central paths are more coexpressed than random paths connecting asthma to other unrelated diseases. To test this hypothesis, we considered the DisGeNet GDA corpus, from which we extracted all the unrelated diseases and phenotypes with number of annotated genes similar to asthma and COPD (between 25 and 35 genes), for a total of 59 phenotypes. We thus measured the SC of random paths connecting the asthma and COPD seed genes to the genes associated to these phenotypes (see Methods). The SC values of the random paths connecting the asthma seeds and each DisGeNet phenotype were measured in the epithelial brushings of asthmatic samples (GSE4302), whereas the SC values between these phenotypes and COPD seeds were measured in the lung tissue of COPD samples (GSE57148). Figure 5a shows the SC distributions of FC paths and random paths of each DisGeNet phenotype in the asthma case (top) and COPD case (bottom). For clarity we show only the distributions

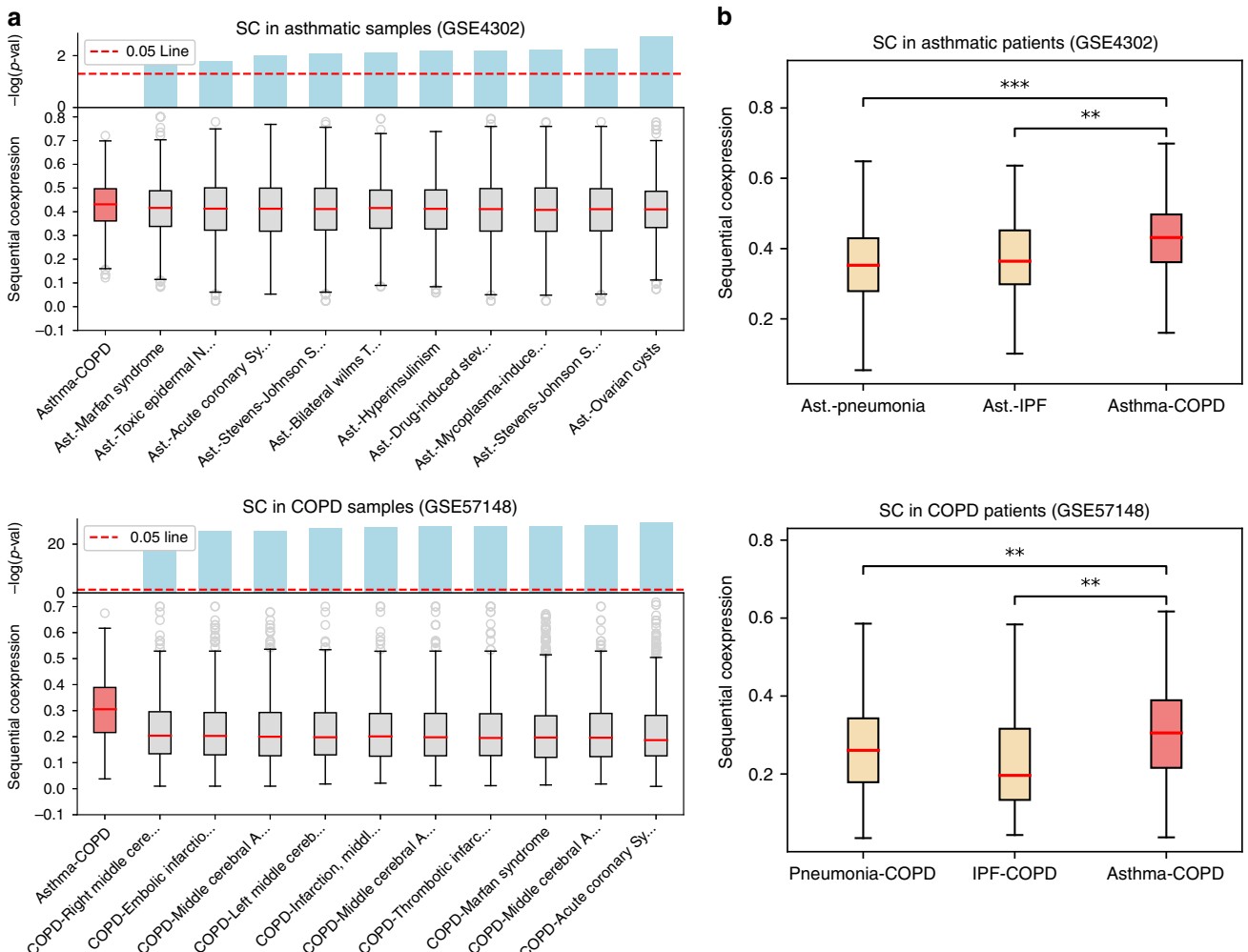

**Fig. 5 Sequential coexpression of random paths connecting to unrelated phenotypes. a** Sequential coexpression (SC) of the FC paths compared with random paths between the asthma module and each DisGeNet phenotype (top) and between the COPD module and each DisGeNet phenotype (bottom). For clarity only the top 10 phenotypes are shown, ordered by increasing significance. **b** (top) SC distribution of the asthma-COPD pair compared with SC of the asthma-pneumonia and asthma-IPF pairs, evaluated on asthmatic samples of GSE4302 data. **b** (bottom) SC distribution of the asthma-COPD pair compared with SC of the COPD-pneumonia and COPD-IPF pairs, evaluated on COPD samples of GSE57148 data. In the boxplots, boxes indicate the quartiles, whiskers extend to an additional 1.5 * IQR interval, and the medians are highlighted in red. One, two, and three asterisks, respectively, denote a Mann–Whitney $p$-value that is <0.05, 1e−4, and 1e−10, and "n.s." stands for a non-significant result.

of the top 10 phenotypes, ordered by their $p$-value scores (bars at the top of each plot). In both cases we find that the FC paths are characterized by significantly larger coexpression values, confirming the close relationship between asthma and COPD. In order to further test the specificity of the asthma-COPD relation and account for eventual intrinsic biases of the processing steps, including disease module construction and flow centrality evaluation, we re-executed the whole processing pipeline between asthma and two related diseases of the lung, pneumonia and idiopathic pulmonary fibrosis (IPF) (see Methods). We find that asthma and COPD are characterized by higher SC values with respect to asthma ↔ pneumonia and asthma ↔ IPF pairs in the epithelial brushings of asthmatic samples (GSE4302) (Fig. 5b, top). We then repeated the same analysis for the pairs COPD-pneumonia and COPD-IPF, obtaining a similar result in lung tissue of COPD samples (GSE57148) (Fig. 5b, bottom). This result suggests that the molecular interaction of asthma and COPD may be deeper than expected when compared with other lung diseases, as conjectured by the Dutch hypothesis.

**Overexpression and knockdown experiments in cell lines.** To further validate the FC approach, we used in vitro gene perturbation to experimentally establish a connection between an asthma source seed gene and a COPD target seed gene via a network path of high flow centrality (see Methods). For this, we focused our attention on the asthma seed gene *GSDMB*, one of several genes on *17q21* that harbors the most replicated asthma-susceptibility locus identified by GWAS[63]. *GSDMB* is expressed in bronchial epithelium (a cell type relevant to the pathogenesis of both asthma and COPD) and recent murine models suggest that *GSDMB* overexpression results in spontaneous airway remodeling[64]—subepithelial fibrosis—that in humans contributes to fixed airway obstruction observed in COPD. For this experiment, we considered all the flow central paths between *GSDMB* and any of the COPD seed genes (Fig. 6a), i.e., those paths where all the intermediate genes have a significant FCS. To maximize the sensitivity of the analysis we consider as significant those genes whose FCS is >2 or whenever the right-tailed empirical $p$-value of their flow centrality value is <0.05. We find 8 paths satisfying

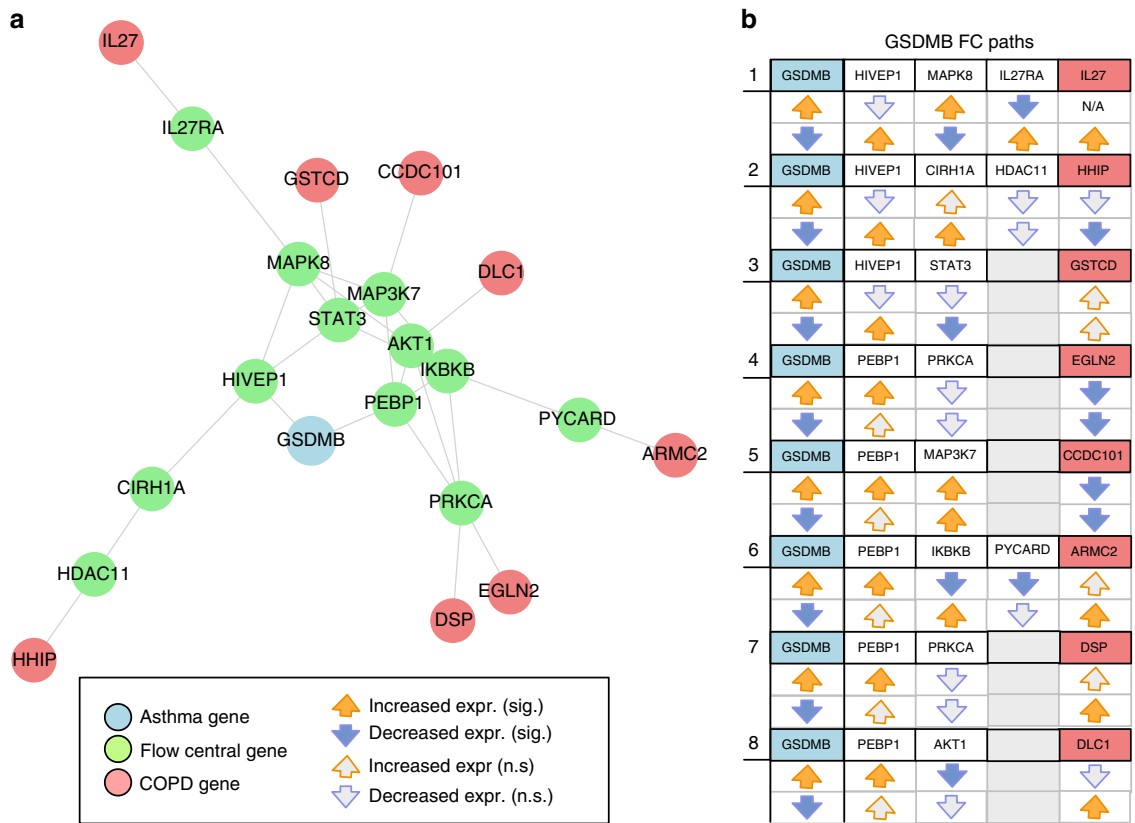

**Fig. 6 Flow central paths between GSDMB and COPD seed genes. a** Subnetwork of the nodes in the FC paths; **b** Set of eight flow central paths with GSDMB as source node and relative downstream changes in expression after overexpression/knockdown of GSDMB. For each column, the colored orange (blue) arrows represent significant overexpression (downregulation) of the corresponding gene, while gray arrows represent non-significant changes.

these criteria. Of note, all eight flow central paths pass through one of two *GSDMB* neighbors *HIVEP1* and *PEBP1* (Fig. 6b). In experiments conducted in triplicate in a human bronchial epithelial cell line, we either augmented or suppressed *GSDMB* mRNA expression by plasmid transfection or siRNA knockdown, respectively, and obtained expression data for *GSDMB*, all predicted flow central genes, and target COPD seed genes from RNA-seq profiles of global gene expression (see Methods for details). We found strong evidence for connections between the asthma seed *GSDMB* and its predicted downstream target COPD seeds *IL27*, *HHIP*, and *GSTCD*. As summarized in Fig. 6b, both overexpression and silencing of *GSMDB* resulted in reciprocal downstream alterations in the expression of most flow central genes and target COPD genes. For example, *GSDMB* silencing resulted in significant changes in the expression of flow central *HIVEP1* (expression increased), *MAPK8* (decreased), *IL27RA* (increased), and the COPD seed gene *IL27* (increased), while *GSDMB* overexpression resulted in changes in expression opposite to those induced by *GSDMB* silencing (*MAPK8* increased, *IL27RA* decreased, with non-significant decreased expression of *HIVEP1*, see path 1 in Fig. 6b. Baseline expression of *IL27* was below meaningful detection levels, precluding its analysis). Similar patterns were observed for most genes in paths connecting *GSDMB* to *HHIP* and *GSTCD*.

## Discussion
The causal relationships of complex diseases are elusive because often multiple mechanistic processes explain why these diseases occur and develop in many different forms. However, with the advent of sequencing technologies and multi-omic assays it is now possible to obtain a more complete overview of the genetic

profiles that are more susceptible to developing a condition. The long-standing question of the potential mechanistic relationships between asthma and COPD can thus be approached from a molecular viewpoint, and the putative causes analyzed at the level of genes and proteins. Yet, the information obtained by such technologies is mostly about the 'actors' of the processes, more than the processes themselves, leaving room for targeted studies analyzing the relations between the genes involved in disease development and pathways cross-talk.

The analysis of protein–protein interactions connecting the two diseases represents a first step in disentangling the intricate pathways that are responsible for the common pathogenesis of diseases such as asthma and COPD.

In this work we defined flow centrality, a topological measure to detect the genes mediating the molecular interactions occurring between asthma and COPD. Flow central genes show high specificity and can not be trivially associated to disease genes through first-neighbor interactions. By analyzing the network paths connecting asthma to COPD, we showed that flow central genes are functionally similar to the seed genes of the two diseases. This pattern is quite general: for a multitude of related disease pairs we observed high functional similarity between the flow central genes and their respective sources and targets, suggesting that flow centrality captures low-level molecular mechanisms that underlie different pathological conditions. As further support of this hypothesis, we measured high coexpression between flow central genes and the disease genes of asthma and COPD in multiple human transcriptomics datasets. To obtain experimental evidence of the regulation patterns occurring between the asthma and COPD genes, we restricted our attention to *GSDMB*, one of the most replicated genes associated to asthma,

and assessed the downstream effects of its perturbation through in vitro overexpression/knockdown experiments. The flow central nodes occurring within the network paths connecting *GSDMB* to the COPD seed genes show strong differential expression patterns, hinting that these genes could participate in the molecular mechanisms carrying the perturbation from the asthma-specific to the COPD-specific domain.

These results suggest that flow centrality can help in identifying the genes involved in the key pathways associated with the transitioning or hybrid phenotypes between the two diseases. Multi-omics measurements (such as transcriptomics, genomics and epigenomics assays) could be leveraged to define a molecular profile of the flow central genes in affected patients[65]. By correlating these molecular profiles with the patients' clinical conditions and outcomes, it would be in principle possible to locate these profiles on the asthma-COPD spectrum, creating new opportunities for targeted therapeutics.

The effectiveness of the flow centrality approach depends on the reliability of current PPI data. However, it is estimated that only around 20% of the total protein interactions are known, and a considerable number of the modeled interactions could be the result of false positive interactions[5]. Moreover, since the discovery of real interactions is nonuniform, and mainly driven by the interest in researching proteins that are associated to important functions or diseases, it may result in an inaccurate modeling of the actual wiring patterns of the network. However, the increase in reliability allowed by new and improved bias-free experimental and prediction[66] assays of protein interactions (such as yeast-two hybrid), will be crucial in refining our understanding of the genes responsible for carrying a disease perturbation.

## Methods

**Construction of the interactome**. The network we utilized in this work has been compiled by Cheng et al.[19], and integrates protein–protein interactions extracted from 15 databases:

1. Binary PPIs tested by high-throughput yeast-two-hybrid (Y2H) systems (refs. [67,68], http://interactome.baderlab.org).
2. Kinase-substrate interactions from KinomeNetworkX[69], Human Protein Resource Database (HPRD)[70], PhosphoNetworks[71,72], PhosphositePlus[73], DbPTM 3.0[74], and Phospho. ELM[75].
3. PPIs identified by affinity purification followed by mass spectrometry (AP-MS), Y2H and by literature-derived low-throughput experiments, and protein three-dimensional structures from BioGRID[76], PINA[77], Instruct[78], HPRD[70], MINT[79], IntAct[80], and InnateDB[81].
4. Signaling network by literature-derived low-throughput experiments as annotated in SignaLink2.0[82].

By considering only the largest connected component of the network and removing self-loops, the resulting interactome includes 16,656 proteins and 243,592 interactions. For further details, refer to ref. [19].

**Asthma and COPD seed genes**. We identified a set of well-established genes by aggregating several sources of genome-wide associations studies that have been replicated for COPD and asthma susceptibility and specific genes implicated by eQTL or functional studies within GWAS regions. The sources considered for asthma and COPD are detailed, respectively, in Supplementary Data 1 and 2. For COPD, we also considered genes causing Mendelian syndromes which include emphysema as part of their phenotypes: alpha-1 antitrypsin deficiency (*SERPINA1*) and cutis laxa (*ELN* and *FBLN5*).

**Disease module construction**. The asthma and COPD disease modules are built through the DIAMOnD algorithm[20]. DIAMOnD is based on an iterative scheme that exploits the network's topology to gradually build a disease module. Given a disease gene set of $N_s$ genes, at each iteration DIAMOnD calculates the statistical significance of connectivity of each node of the network to the disease genes. If the disease module at the current iteration is composed of $s$ genes, then a candidate node with degree $k$ and $k_s$ edges connected to the $s$ genes in the module has a $p$-value

$$p\text{-val}(k, k_s) = \sum_{k_i = k_s}^{k} p(k, k_i) \tag{1}$$

where $p(k, k_i)$ is the hypergeometric distribution

$$p(k, k_i) = \frac{\binom{s}{k_s}\binom{N-s}{K-K_s}}{\binom{N}{k}} \tag{2}$$

and $N$ is the total number of genes in the network. In ref. [20], seed genes can be weighted in order to be more preponderant in the $p$-value calculation, but in this analysis this possibility is not explored. Among the candidate nodes, the node that is most significantly connected to the set (and thus has a smaller $p$-value) is added to the module and the procedure starts again with the increased gene set. This operation is repeated for a fixed number of iterations $N$, reaching a final module size of $N_s + N$ genes. In order to choose $N$ we extracted from UK-Biobank[21] the genes significantly associated with asthma and COPD, using a threshold $p$-value of $1e-3$, and not present, respectively, in the asthma and COPD seed genes set. While UKB genes are in general different from the seed genes of asthma and COPD, some overlap may occur. Therefore, we considered only the UKB genes that are not present in the seed genes of asthma and COPD, respectively, 742 and 458 genes. Starting from the asthma seed genes we executed DIAMOnD and, at each iteration, we measured the hypergeometric $p$-value between GWAS-significant genes and the genes in the current module, obtaining the curve shown in Supplementary Fig. 1(a, left). We then selected as iterations cutoff $N$ the value that yielded the lowest $p$-value in the curve. We repeated the same operations for the COPD module (Supplementary Fig. 1(b, right). The final sizes of the asthma and COPD modules are, respectively, 373 genes and 228 genes, with 14 overlapping genes.

**Significance of overlap between the modules**. In order to test the significance of the overlap between the asthma and COPD modules we generated 1000 random pairs of gene sets of asthma and COPD with the procedure described below (section Gene set randomization in Methods), and calculated the fraction of times when the measured overlap between random samples is equal or greater than the observed value (14 genes).

**Flow centrality**. Given a source disease module $T$ and a destination module $S$, we define the flow centrality of a node $v$ is given by

$$\text{FC}^{S,T}(v) = \frac{1}{|S||T|} \sum_{s \in S, t \in T} \frac{\sigma_{st}(v)}{\sigma_{st}} \tag{3}$$

where $\sigma_{st}(v)$ is the number of shortest paths from $s$ to $t$ passing through node $v$, $\sigma_{st}$ is the total number of shortest paths between $s$ and $t$, and $|\cdot|$ is the size of the corresponding set. In the particular case when $S = T = V$, where $V$ is equal to the set of all the nodes of the networks, then the flow centrality reduces to the betweenness centrality measure. Note that, while Eq. (3) implies a directionality between the source disease module $S$ and target module $T$, in undirected networks such roles are interchangeable.

The raw values of flow centrality as calculated by Eq. (3) are biased toward hubs: high-degree nodes are more likely to participate in shortest paths between node pairs just by chance. To account for this bias we calculate the statistical significance of the obtained values by comparing them with a null distribution generated by randomizing 1000 times the source and target modules. The details of the randomization scheme are described in Methods section. For each random pair of source and target modules we calculate the flow centrality of each node of the network and measure the average $\mu_{\text{FC}}$ and standard deviation $\sigma_{\text{FC}}$ across all the samples. The FCS of a node $v$ is then calculated as

$$\text{FCS}^{S,T}(v) = \frac{\text{FC}^{S,T}(v) - \mu_{\text{FC}}}{\sigma_{\text{FC}}}. \tag{4}$$

A large positive FCS indicates that the node is more likely to occur in the shortest paths connecting the source and target modules, while a small or negative value suggests that the node is not relevant to the chosen pair of modules.

**FCS stability**. To evaluate the stability of FCS values to moderate variations of the boundaries of the disease modules we performed the following test. We defined a range of possible small variations in the selected cutoff value iteration of DIA-MOnD modules, i.e., $\Delta \in \{-30, -20, -10, -5, -1, 1, 5, 10, 20, 30\}$. For example, when considering a variation $-30$ from the list in the case of the asthma module (373 genes), we build a perturbed asthma module by considering only the first $N - 30$ genes prioritized by DIAMOnD, where $N$ is the original cutoff value, obtaining a module size of $N_{\text{asthma}} - 30 = 343$ genes. We repeat the same scheme for COPD. For each value of $\Delta$ we calculate the perturbed FCS values by setting the perturbed modules as source and target. The perturbed FCS are then compared with the original ones (see Supplementary Fig. 4), and Supplementary Fig. 5 shows their Spearman's correlation for each value of $\Delta$. The obtained correlation values are very high ($\geq 0.94$), indicating the robustness of the FCS scores to moderate variations of the modules size.

**Gene set randomization**. We defined a randomization scheme designed to create a null distribution of random modules that are topologically similar to a given

DIAMOnD module. A straightforward way to generate randomized genes sets would be to select a number $N$ of random genes in a degree-preserved way, where $N$ is the size of the disease module we want to randomize, and repeat this process a number of times to obtain the samples. This method, however, has the drawback of generating disease modules that are quite different from the asthma and COPD sets we calculated with DIAMOnD. DIAMOnD iteratively searches in the neighborhood of the seed genes, generating modules that are more compact and well interconnected with respect to a random selection. Therefore, a $z$-score evaluated by comparing on such samples would be confounded by the different topological properties of the random modules. For this reason we defined the following randomization scheme:

1. Given a set of $N_{seed}$ seed genes of a disease module $\mathcal{M}$ (obtained with DIAMOnD), we extract a new set of $N_{seed}$ random seed genes in a degree-preserved way.
2. We run DIAMOnD on the set of random seed genes for $N$ iterations, where $N$ is the size of $\mathcal{M}$, obtaining a new random module of size $N$.

In this way, the procedure generates random modules that are topologically more similar to those generated by DIAMOnD.

**Selection of network paths**. The flow central paths are selected as all the shortest paths connecting the asthma and COPD seed genes, whose intermediate genes (i.e., those genes that are not the source or destination of the path) have a flow centrality score of 2 or greater. Assuming a normality in the null distribution of the FC values, a value that is 2 standard deviations away from the average value is well outside the bulk of the null distribution. Choosing an excessively large threshold can result in too few nodes being selected and might lead to missing important nodes in areas of lower edge density, while a too low threshold would increase the false positives. As an additional constraint we require for all the intermediate nodes in the paths to participate in at least five shortest paths connecting COPD and asthma nodes, in order to remove from the pool all the nodes that have unstable FCS values because of low shortest-path statistics. Note that while the full disease module information has not been used to select the initial pool of shortest paths, this information is embedded in the calculation of flow centrality of each gene in the network, since the FC depends on the source and target disease modules.

The Type A path randomization scheme is structured as follows:

1. Extract one length value $L$ from the empirical distribution of FC path lengths.
2. Create an empty path $\mathcal{P}$.
3. Select a node $n$ uniformly at random in the network and add it to $\mathcal{P}$.
4. Select one random neighbor of $n$ among those not already in $\mathcal{P}$ and add it to $\mathcal{P}$.
5. Repeat from step 3 until the length of $\mathcal{P}$ is $L$.
6. Add $\mathcal{P}$ to the current set of random paths.
7. Repeat from step 1 until a desired number of random paths is obtained.

Note that in the actual implementation of the scheme above some additional controls are performed in order to account for edge cases such as when no new neighbors can be added to the path, etc.

The Type B random paths are selected by uniformly sampling paths from the pool of shortest paths connecting the genes of the two diseases.

**Sequential similarity**. Given a path $\mathcal{P}^{(n)}$ of length $n$ as an ordered sequence of unique genes in the network $(g_1, g_2, g_3, ..., g_n)$. The sequential similarity is then defined as

$$s_{seq}(\mathcal{P}^{(n)}) = \frac{1}{n-1} \sum_{i=1}^{n-1} s(g_i, g_{i+1}) \tag{5}$$

where $s(\cdot, \cdot)$ is any GO terms similarity measure between genes. In this work we considered the best-match average (BMA) of Resnik's similarity measure[83,84], defined as follows. Given two genes $u$ and $v$ associated to the sets of GO terms $\mathcal{U}$ and $\mathcal{V}$, respectively, the BMA Resnik similarity has the form

$$s(u, v) = \frac{1}{|\mathcal{U}| + |\mathcal{V}|} \left[ \sum_{\alpha \in \mathcal{U}} \max_{\beta \in \mathcal{V}} [sim(\alpha, \beta)] + \sum_{\beta \in \mathcal{V}} \max_{\alpha \in \mathcal{U}} [sim(\alpha, \beta)] \right]$$

where $sim(\alpha, \beta)$ denotes the Resnik similarity measure between the GO terms $\alpha$ and $\beta$.

**Sequential coexpression**. Given a path $\mathcal{P}^{(n)}$ of length $n$ as an ordered sequence of unique genes in the network $(g_1, g_2, g_3, ..., g_n)$. The sequential coexpression is then defined as

$$\rho_{seq}(\mathcal{P}^{(n)}) = \frac{1}{n-1} \sum_{i=1}^{n-1} \left| \rho\left(\mathbf{e}_{g_i}, \mathbf{e}_{g_{i+1}}\right) \right| \tag{7}$$

where $\mathbf{e}_g$ is the random variable indicating the expression values of gene $g$ and $\rho(\cdot, \cdot)$ is the Pearson correlation. The sequential coexpression is the absolute correlation of the expression of adjacent genes in a network path, and therefore it

measures the extent of coordination in the gene expression along the path. Notice that multiple transcripts in the expression data can be associated to the same gene. In those cases, we calculated the sequential coexpression as the maximum absolute value of correlation between all the possible pairs of probes/transcripts associated to the two genes. If at least one gene is not present in the expression dataset considered, then the sequential coexpression of path $\mathcal{P}^{(n)}$ is considered null and excluded from the analysis.

**Sequential similarity of related disease pairs**. We downloaded from the DisGeNet repository[50] all the curated gene-disease associations (GDA) and the disease mappings to convert the UMLS CUI identifiers to the identifiers of several other vocabularies. Note that in order to limit the amount of false positives, the DisGeNet associations obtained by text mining of MEDLINE abstracts (extracted through the BeFree tool) are excluded from the analysis. The data have been downloaded on July 19th from the webpage http://www.disgenet.org/downloads. From the GDA data we selected only the annotations to phenotypes of the type "Disease or Syndrome". We then filtered all the diseases that are associated to <50 genes that can be mapped on the PPI network. Each resulting disease is associated to a list of Disease Ontology IDs (DOID), as annotated in the disease mapping data. Disease Ontology is a standardized ontology of human diseases that semantically integrates disease and medical vocabularies through cross mapping and integration of MeSH, ICD, NCI's thesaurus, SNOMED CT and OMIM[85,86]. Notice that a disease can be mapped to multiple DOIDs, since it can belong to multiple categories in the ontology tree. We proceeded to calculate the pairwise similarities between diseases, using the R package DOSE[87]. For each pair, the similarity is calculated as the maximum Resnik similarity between all their associated DOIDs. The calculated similarities are shown in Supplementary Fig. 2, where the similarities that could not be retrieved (i.e., returned as null by the DOSE function) are set as 0. The related disease pairs are selected as those pairs with:

- Similarity greater than the 90th percentile in the overall distribution of similarities, not considering the similarities that could not be retrieved and the similarities equal to 0.
- Similarity <1, to avoid selecting disease IDs that are synonyms of the same phenotype.
- Number of overlapping associated genes <10, to retrieve related diseases with little common genetic basis, as in the asthma-COPD case.

After applying this criteria, the resulting disease pairs are 66, listed in Supplementary Data 5. We manually scrutinized the disease pairs to assess the existence of an actual relation between them, obtaining for most of them a positive match. Given a disease pair D1–D2, we evaluated the flow centrality values of all the nodes in the network, by following the scheme outlined in the main text, with the only difference being in the generation of the random modules for the FCS calculation. In this case the disease modules correspond to the set of GDA retrieved from DisGeNet, without recurring to DIAMOND prioritization, and thus the random samples are obtained with a simple degree-preserved randomization of the disease genes. After the FCS values are calculated, we selected the corresponding FC paths, extracted the random paths of Type A (RdmA) and B (RdmB) as described in the main text, and evaluated the three distribution of sequential similarities (SS). For each disease pair we perform two right-sided Mann–Whitney tests, comparing the SS of the FC paths with the SS of RdmA and the SS of RdmB, obtaining two $p$-values $p_A$ and $p_B$. A final $p$-value is calculated as $\max(p_A, p_B)$. Significance of the aggregated $p$-value implies that the SS of the FC paths are significantly greater than the SS of both RdmA and RdmB, and thus the FC paths are more likely to represent meaningful biological links between the two diseases.

To assess the specificity of the result, for each disease pair D1–D2 in the pool defined above we generated two sets of 100 random modules, by randomizing the disease genes of D1 and D2 in a degree-preserving way. By using the values of flow centrality evaluated for the original pair, we selected the flow central paths of each random pair, i.e., the shortest paths connecting the nodes of the two random modules where all the intermediate genes have flow centrality >2. We refer to these paths as random FC paths. Then, we performed a Mann–Whitney test between the distribution of SS values of the original disease pair and the SS values of each random pair, separately. As a result of this operation we obtained 100 $p$-values comparing the SS of the actual disease pair with the SS of each random pair.

**Selection of random diseases and phenotypes**. In order to test the significance of the sequential coexpression of the flow central paths, we considered a number of diseases and phenotypes from the DisGeNet repository[50] that are unrelated to asthma and COPD. The objective of this test is to compare the coexpression of the FC paths of asthma and COPD with random paths connecting asthma to a random disease, and repeat the same for COPD. We selected only the diseases and phenotypes with gene set sizes similar to the asthma and COPD seed gene sets, i.e., between 25 and 35 genes after mapping to the PPI network. In addition, we restrict the selection to the phenotypes annotated as "Disease or Syndrome". With this criterion we obtained a total of 59 diseases and phenotypes. For each of these gene sets, we sampled 10,000 shortest paths by iteratively choosing one random seed gene of asthma as source and one random gene in the set as target, and repeated the same for COPD.

**GEO expression data**. We considered 18 microarray and RNA-seq expression datasets from GEO, as detailed in Supplementary Data 6. For each dataset, standard data processing steps were applied, such as conversion of probe IDs and gene symbols to entrez IDs and log-transformation, when necessary. For each expression dataset different subgroups of samples were selected, depending on the information available. Samples were first divided in disease or healthy condition, when present, and analyzed separately. Different tissues or cell types in the same dataset where further divided in separate classes and analyzed separately, when the information were available. For example, in GSE104468 data the asthmatic and control samples are further divided in bronchial epithelia, nasal epithelia and PBMC. Classes of samples that were not relevant for the analysis of the asthma-COPD overlap were excluded. For example, since allergy is an asthma-specific feature, atopic samples were excluded from the analysis when non-atopic counterparts were available (e.g., GSE473). In addition, in some cases we also considered overlapping groupings. For example, in GSE37147, we selected both the class of COPD smokers and its subclass of COPD smokers with no history of asthma. More details on the selected classes and the corresponding numbers of samples are provided in Supplementary Data 6.

**SC of asthma and COPD with pneumonia and IPF**. We selected the seed genes of pneumonia and idiopathic pulmonary fibrosis (IPF) from the DisGeNet repository, obtaining, respectively, 52 and 18 genes mapped on the PPI. In order to build a module with the same size as the COPD module, we run DIAMOnD with $N = N_{COPD} - 52 = 176$ iterations for pneumonia and $N = N_{COPD} - 18 = 210$ iterations for IPF, where $N_{COPD} = 228$ is the size of the COPD module. We then evaluated the flow centrality of the PPI nodes with the asthma module as source and the pneumonia module as target gene set, and repeat the same for asthma and IPF. The FCS of each gene is calculated by randomizing the asthma, pneumonia and IPF modules with the procedure described above. In brief, seed genes are randomized in a degree-preserved way, and DIAMOnD is subsequently executed on the random gene sets to create the random modules. The sequential coexpression of the two pairs is then evaluated and compared with the SC of the asthma-COPD pair on expression data of asthmatic patients (GSE4302). The same process is repeated for COPD-pneumonia and COPD-IPF on expression data of COPD patients (GSE57148).

**Overexpression and knockdown experiments**. Cell culture: Human bronchial epithelial cell line Beas-2B or 16HBE cells were purchased from ATCC and cultured in Dulbecco's modified Eagle medium (DMEM) or Eagle's minimal essential medium (EMEM), respectively, supplemented with 10% fetal bovine serum, penicillin (50 units/ml), and streptomycin (50 g/ml).

Overexpression of recombinant *GSDMB* in Beas-2B cells: Human *GSDMB* in pCMV6 (epitope-tagged with Myc and FLAG, both at the carboxy-terminus) purchased from Origene (catalog number RC202279). Beas-2B cells were plated in 6-well plates at $4 \times 10^5$ cells/well overnight in complete medium. The next day, 0.5 μg of *GSDMB* plasmids or control plasmids and 1 μl of Lipofectamine 3000® (Thermo Fisher) were added into cells with fresh DMEM medium according to the manufacturer's instructions in triplicate wells. RNA extraction was done at 48 h after transfection for RNA sequence.

siRNA knockdown in 16HBE cells: 16HBE cells were plated in 6-well plates at $6 \times 10^5$ cells/well overnight in complete medium. The next day, 30 pmol of *GSDMB* siRNA or control siRNA and 5 μl of Lipofectamine RNAiMAX (Thermo Fisher) were added into cells with fresh EMEM medium according to the manufacturer's instructions in triplicate wells. Two different hairpins (Thermo Fisher, s31709, s31711) were used in the experiments. RNA extraction was done at 48 h after transfection for RNA sequence.

**Reporting summary**. Further information on research design is available in the Nature Research Reporting Summary linked to this article.

## Data availability

The authors declare that the main data supporting the findings of this study are available within the article and its Supplementary Information files. Extra data are available from the corresponding author upon request.

## Code availability

The source code for reproducing the analysis has been developed in python 3.6 and is available as a github repository at the url https://github.com/reemagit/flowcentrality.

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

## Acknowledgements

The authors would like to thank Istvan Kovacs, Marc Santolini, and Ayse Kilic for useful discussion, and the Rivas lab for making the Global Biobank Engine resource available. We acknowledge the support of the National Institutes of Health (NIH) grants R01 HL118455-04-1 and P01 HL13285. The funders had no role in study design, data collection, and analysis, decision to publish, or preparation of the paper.

## Author contributions

E.M. and A.S. conceived and developed the idea. E.M. analyzed the data and was the lead writer of the manuscript. F.G., P.H.K. and X.Z. carried out the experiments. S.H.B., E.K.S., A.-L.B., S.T.W., B.A.R. and A.S. contributed to the writing of the paper, provided critical feedback and helped shape the research and the analysis of the problem.

## Competing interests

A.L.B. is founder of Nomix, Foodome and Scipher Medicine, companies that explore the role of networks and food in health. The remaining authors declare no competing interests.
