## [Peer Review File · Nature Communications]

Reviewers' comments:

Reviewer #1 (Remarks to the Author):

This manuscript describes a network-based framework to identify proteins mediating the interaction between asthma and COPD in a protein interaction network. Conceptually, the flow centrality measure is very similar to betweenness centrality measure although the authors mention that it is "variant" of the betweenness centrality". It is not clear though what the difference is.

No formal validation or evaluation of the analysis methods on other diseases is presented. Hence, it is difficult to perceive how relevant this method would be for other pairs of diseases. In other words, while the authors on one hand propose this as a general framework to identify mediating proteins between any two diseases, the analysis is based on just two diseases (asthma and COPD).

It is not clear whether the genes identified as participating in asthma- and COPD-associated biological processes could not have been found by general betweenness centrality measures or through shortest paths between asthma and COPD GWAS genes. In other words, could these genes would not have been found through existing network based approaches (centrality measures, shortest paths, label extensions, or measures based solely on random walk)? The authors should discuss as to how this approach is different from these or specifically other flow centrality measures? For example, Kivimaki et al., 2016 introduced two betweenness centrality measures based on the randomized shortest paths (RSP). Focusing on the shortest paths and also taking into account longer paths, their framework defined Boltzmann probability distributions over paths of the network. One of their measures (RSP betweenness centrality) counts the expected number of visits to a node while the second one (RSP net betweenness) is based on the overall net flow over edges connected to a node. In summary, without a comparison and without a general context, this reviewer has concerns as to how applicable this approach is for other diseases.

The rationale behind selecting only one expression data each for asthma and COPD is surprising. What is the rationale for selecting GSE4302 for asthma when there are several other studies including RNA-seq (e.g., GSE104472, GSE64913, GSE89809, GSE85568, etc.) and some of them have much larger number of disease and control samples? Likewise for COPD study too (e.g., GSE57148). The authors should consider using additional datasets as part of the sequential coexpression. At least, they should consider checking the GSDMB FC paths in additional data sets as part of reproducibility and robustness. Additionally, in the COPD data set that the authors used, there seem to be patients with history of asthma. Were these excluded from the analysis? If not these should be excluded before calculating the sequential coexpression.

While the comparison between asthma-COPD and asthma-pneumonia is interesting, the authors should consider testing other chronic respiratory conditions such as IPF.

Lastly, the patterns of airway inflammation in COPD and COPD are quite different. For instance, in asthma airway inflammation is characterized mostly by activation of mast cells, eosinophil infiltration driven by activation of type 2 cells (T-helper and innate lymphoid cells) while in COPD, mast cell activation is usually not present and there is infiltration of macrophages and neutrophils. It is not very clear if the FC approach will be able to define the overlapping and transitioning/connecting phenotypes and especially if it would be of help in identifying them in the clinic or lead to better management of the patients with asthma and COPD.

The authors can chose to ignore this suggestion. They should probably consider performing this analysis on a global scale in an unbiased way – using GWAS genes – and find out potentially novel FC paths between diseases as a prelude to the current asthma-COPD study (can be a case study).

Reviewer #2 (Remarks to the Author):

In this manuscript by Maiorino and colleagues, the authors employ a novel network-based framework, flow centrality, using the UK Biobank GWAS data, to characterize the interactome (e.g., protein-protein interaction) common to asthma and COPD and identify mediators (e.g., disease-disease interactions) between the two diseases. Flow centrality measures the number of network paths connecting two regions (source and target) that pass through a given node. To demonstrate that genes identified through flow centrality are biologically meaningful, they present results from several steps, including an assessment of potential functional relations with the two diseases using Gene Ontology similarity tests, gene coexpression analysis in human lung cell line transcriptomic data from asthmatics and subjects with COPD, and in vitro genetic perturbation in a bronchial epithelial cell line on the GSDMB gene.

This manuscript summarizes an elegant approach towards the elucidation of networks of molecular interactions beyond the simplistic concept of genes or proteins working in isolation. Most importantly, it underscores the importance of considering candidate 'genes' and resulting proteins as not simply those that are part of the 'disease module', but rather mediators that link pairs of complex diseases. This is highly relevant because pleiotropy is especially characteristic of asthma but the precise underlying mechanisms are to date scarcely understood. Importantly, results from this study not only provide novel, shared molecular mechanisms for asthma and COPD, but are a strong proof of concept of the value of flow centrality in disentangling overlap of other complex traits.

One concern is the premise of the comparison of asthma and COPD. Specifically, the authors justify the overall approach of comparing asthma and COPD in large part on the argument that asthma leads to COPD. While it is true that some studies have shown an increased risk of developing COPD later in life if the individual had severe asthma in childhood (as high as 32 times more likely to develop compared to other types of asthmatics), there is not broad consensus on this hypothesis and indeed there are substantial distinct differences between the two traits (i.e., asthma is generally reversible versus irreversible airflow obstruction, source of inflammation (allergies vs bacteria), response to anti-inflammatory therapeutics, etc). If the goal of the study was to identify overlap between severe childhood asthma that becomes COPD later in life, it is questionable to what extent the GWAS dataset used for this study (the UK Biobank) is ideally suited to test the hypothesis that asthma leads to COPD. That said, the authors present a reasonably comprehensive introduction (pg 2) regarding other overlapping features between asthma and COPD that justifies their overall approach, and the proof of concept studies of GSMB are especially compelling. Despite these concerns and questions below, overall this is a clearly written and reasonably summarized manuscript employing a novel framework for disentangling complex traits.

Major:

1. Further to the concern described above regarding features of overlap between asthma and COPD, did the authors consider to what extent genes identified in this study fall under categories supporting purported common features (i.e., airflow obstruction reversibility) and to what extent genes that represent distinct differences (i.e., allergic disease) did not show up?
2. GWAS database(s) - The group relied on publicly available GWAS data from the UK Biobank as the first step (compiling known GWAS loci, or sets of 'seed' genes, for each asthma and COPD). Can they speculate to what extent findings may have differed had they focused on GWAS data from studies specifically designed to identify genetic determinants associated with asthma and COPD? Specifically, since the most compelling data supporting the argument that asthma leads to COPD is for severe asthma during childhood, and because the authors chose to focus on an asthma candidate gene as a proof of concept which has in fact been specifically associated with childhood onset asthma (GSDMB), did they consider, at a minimum, replication of this approach in other GWAS datasets?

Minor:

1. The statement on pg 2 "people affected by asthma since birth are more likely to develop COPD at later ages" is first of all, not fully substantiated and second, needs referencing (could use references provided on pg 5 for a similar statement although better references could be selected).
2. Figure 2a would be more informative if the actual values were placed over the bars.
3. Figure 3: what was the process for selecting the 3 examples? Was this random, or biased in some way?
4. Table S1 is not readable as is.

We would like to thank the reviewers for their valuable feedback. Following their suggestions, we implemented additional tests to validate and generalize our approach. The outcome resulted in stronger evidence for the effectiveness and robustness of our approach. The changes to the original work also motivated us to re-implement the analyses from scratch, in order to make it more easily reproducible and well organized. The code is now available at the github page <https://github.com/reemagit/flowcentrality>, and can be executed through a makefile.

Additionally, we chose to make small changes in the original analysis and regenerated the random samples with fixed random seed for reproducibility. Thus, while the general conclusions remain the same, some quantities change slightly from the previous version. All the details of the changes are reported at the end of this document and in response to the reviewers, when relevant. The sections in the manuscript that underwent major modifications or additions are highlighted in red in the marked manuscript file.

We are grateful to the reviewers for helping us to significantly improve our work and for motivating us to explore further the capabilities and limits of our approach.

Reviewers' comments:

Reviewer #1 (Remarks to the Author):

1.1. This manuscript describes a network-based framework to identify proteins mediating the interaction between asthma and COPD in a protein interaction network. Conceptually, the flow centrality measure is very similar to betweenness centrality measure although the authors mention that it is “variant” of the betweenness centrality”. It is not clear though what the difference is.

The difference between flow centrality and betweenness centrality lies in the set of source-destination nodes that are considered when calculating the flows for each node. In betweenness centrality the sum is over all the nodes of the network (both for source and target nodes), while in the flow centrality formula only a specific set of source nodes and a specific set of target nodes are considered. The practical difference is that the betweenness centrality quantifies the general centrality of a node with respect to any position of the network, while flow centrality quantifies the centrality of a node with respect to two specific source and target subnetworks.

We clarified the difference between flow centrality and betweenness centrality in the manuscript (Section 2.1), and this point is further discussed in question 3 of reviewer 1 below.

1.2. No formal validation or evaluation of the analysis methods on other diseases is

presented. Hence, it is difficult to perceive how relevant this method would be for other pairs of diseases. In other words, while the authors on one hand propose this as a general framework to identify mediating proteins between any two diseases, the analysis is based on just two diseases (asthma and COPD).

Thanks to the reviewer's feedback we had the opportunity to find stronger evidence on the generalization capabilities of our approach.

We performed several tests to validate the methods on other pairs of diseases. The new analysis results are included in section 2.2.2 (Gene ontology similarity of flow central paths between related diseases) and Material and Methods Sec. 4.10. In brief, we considered the corpus of diseases included in DisGeNet annotations, filtering out all the diseases with less than 50 annotated genes. We adopted this criterion because for diseases with a sufficient number of annotated genes it is not necessary to run an additional disease module construction step, since their shortest path statistics will be sufficient to calculate a reliable FC score. We then calculated their pairwise similarities according to their annotations to Disease Ontology terms. From the matrix of similarities we extracted the pairs of diseases with the top 10% similarity values and for which their overlap is less than 10 genes (66 pairs), in order to reduce to those cases where two related diseases have little genetic overlap, as in the asthma/COPD case. Analogously as the asthma/COPD case, for each disease pair we calculated the sequential similarities of their Flow Central (FC) paths and the corresponding random paths of Type A and B. When comparing the distributions, in the majority of cases we observed a statistically larger sequential similarity in the FC compared to the random samples, indicating genes involved in the FC paths are more likely to be functionally related (Fig. 2c).

We also assessed the specificity of this result. For each disease pair, we randomized the disease genes of the source and destination sets 100 times, obtaining 100 random disease pairs. We then selected their corresponding FC paths and evaluated the distribution of their sequential similarities (SS). We observed that the SS of the FC paths are significantly larger than their random counterparts for almost every random realization and in almost every disease pair (Fig. S5), as quantified by the Mann-Whitney one-tailed test.

Together, these results suggest that for a large variety of diseases flow centrality highlights biologically relevant sets of nodes that are functionally related.

1.3. It is not clear whether the genes identified as participating in asthma- and COPD-associated biological processes could not have been found by general betweenness centrality measures or through shortest paths between asthma and COPD GWAS genes. In other words, could these genes would not have been found through existing network based approaches (centrality measures, shortest paths, label extensions, or measures based solely on random

walk)? The authors should discuss as to how this approach is different from these or specifically other flow centrality measures? For example, Kivimaki et al., 2016 introduced two betweenness centrality measures based on the randomized shortest paths (RSP). Focusing on the shortest paths and also taking into account longer paths, their framework defined Boltzmann probability distributions over paths of the network. One of their measures (RSP betweenness centrality) counts the expected number of visits to a node while the second one (RSP net betweenness) is based on the overall net flow over edges connected to a node. In summary, without a comparison and without a general context, this reviewer has concerns as to how applicable this approach is for other diseases.

As we mentioned in response to question 1, the fundamental difference between flow centrality and betweenness centrality lies in FC's dependence on a specific source and target gene sets whereas the classic betweenness measures and also the RSP-based betweenness proposed in Kivimaki et al., 2016 are focused on identifying genes that are central with respect to the entire network. This implies that the genes identified by any of these measures will be nonspecific by definition, since they depend only on the global network topology, and they will not change when a different disease-disease pair is considered.

While most of these measures, RSP betweenness included, could be easily modified to account for restricted source and target node sets, to our knowledge this possibility is almost unexplored in literature. For example, in the source code for the RSP betweenness measure (<https://github.com/ikivimak/RSP-betweenness>) there is no option to calculate the betweenness values for restricted sets of source and target genes.

Another important feature of our measure is the randomization of source and target gene sets, which allows us to calculate a score that is not affected by degree bias.

In Fig. S2 we show the relation between several betweenness measures: flow centrality score (FCS), degree, betweenness, current flow betweenness proposed in Newman 2005 (RW betweenness), and RSP betweenness for three settings of temperature values. As shown in figure S3, all the measures are highly correlated with degree centrality, except for FCS. This implies that FCS is not strongly influenced by the degree and is tailored to a specific set of source and target sets of nodes of interest, in this case the asthma and COPD disease modules.

We added these considerations to the main manuscript in section 2.1 (Flow centrality between modules).

1.4. The rationale behind selecting only one expression data each for asthma and COPD is surprising. What is the rationale for selecting GSE4302 for asthma when there are several other studies including RNA-seq (e.g., GSE104472, GSE64913, GSE89809, GSE85568, etc.) and

some of them have much larger number of disease and control samples? Likewise for COPD study too (e.g., GSE57148). The authors should consider using additional datasets as part of the sequential coexpression. At least, they should consider checking the GSDMB FC paths in additional data sets as part of reproducibility and robustness. Additionally, in the COPD data set that the authors used, there seem to be patients with history of asthma. Were these excluded from the analysis? If not these should be excluded before calculating the sequential coexpression.

As suggested by the reviewer, we extended the analysis to several other datasets collected from GEO. Specifically, we repeated the same analysis on all the datasets suggested by the reviewer (GSE104472, GSE64913, GSE89809, GSE85568, GSE57148) plus several other asthma and COPD expression datasets analyzed in (Sharma et al., Hum.Mol.Gen, 2015) and (Sharma et al., Sci.Rep., 2018), for a total of 18 datasets (including the original ones). We chose different stratifications of the samples whenever more information on the cell type, tissues or conditions were available. All the details are summarized in Table S6. We observe that we can reproduce our results in most of these datasets (13 datasets out of 18), obtaining significant differences between the sequential co-expression values of FC paths compared to random paths of Type A and B. After the reimplementing of the analysis, we noticed an error in the evaluation of the p-values for the GSE37147 (COPD) and GSE4302 data (asthma). The new p-values have been corrected. While we still find high significance in GSE4302, the differences in GSE37147 are no longer significant. All the processing steps can be verified in the source code at the github page <https://github.com/reemagit/flowcentrality>

1.5. While the comparison between asthma-COPD and asthma-pneumonia is interesting, the authors should consider testing other chronic respiratory conditions such as IPF.

We thank the reviewer for the suggestion. We repeated the analysis for the pair asthma-IPF on GSE4302 and for pneumonia-COPD and IPF-COPD on GSE57148, obtaining that FC paths of the asthma-COPD pair are enriched in higher sequential coexpression values with respect to FC paths of asthma-IPF and FC paths of IPF-COPD pairs. This result provides further evidence on the existence of a genetic relationship between asthma and COPD.

1.6. Lastly, the patterns of airway inflammation in COPD and COPD are quite different. For instance, in asthma airway inflammation is characterized mostly by activation of mast cells, eosinophil infiltration driven by activation of type 2 cells (T-helper and innate lymphoid cells) while in COPD, mast cell activation is usually not present and there is infiltration of macrophages and neutrophils. It is not very clear if the FC approach will be able to define the overlapping and transitioning/connecting phenotypes and especially if it would be of help in

identifying them in the clinic or lead to better management of the patients with asthma and COPD.

The authors can chose to ignore this suggestion. They should probably consider performing this analysis on a global scale in an unbiased way – using GWAS genes – and find out potentially novel FC paths between diseases as a prelude to the current asthma-COPD study (can be a case study).

We thank the reviewer for the suggestion and we hope that the implemented tests and analyses are found to be convincing. Since we consider the asthma-COPD overlap case as a unique problem and the main subject of our work, we chose to keep the general focus of the paper on asthma-COPD, and implemented the validation on other diseases as a part of section 2.2.1.

Reviewer #2 (Remarks to the Author):

2.1. In this manuscript by Maiorino and colleagues, the authors employ a novel network-based framework, flow centrality, using the UK Biobank GWAS data, to characterize the interactome (e.g., protein-protein interaction) common to asthma and COPD and identify mediators (e.g., disease-disease interactions) between the two diseases. Flow centrality measures the number of network paths connecting two regions (source and target) that pass through a given node. To demonstrate that genes identified through flow centrality are biologically meaningful, they present results from several steps, including an assessment of potential functional relations with the two diseases using Gene Ontology similarity tests, gene coexpression analysis in human lung cell line transcriptomic data from asthmatics and subjects with COPD, and in vitro genetic perturbation in a bronchial epithelial cell line on the GSDMB gene.

This manuscript summarizes an elegant approach towards the elucidation of networks of molecular interactions beyond the simplistic concept of genes or proteins working in isolation. Most importantly, it underscores the importance of considering candidate ‘genes’ and resulting proteins as not simply those that are part of the ‘disease module’, but rather mediators that link pairs of complex diseases. This is highly relevant because pleiotropy is especially characteristic of asthma but the precise underlying mechanisms are to date scarcely understood. Importantly, results from this study not only provide novel, shared molecular mechanisms for asthma and COPD, but are a strong proof of concept of the value of flow centrality in disentangling overlap of other complex traits.

One concern is the premise of the comparison of asthma and COPD. Specifically, the authors justify the overall approach of comparing asthma and COPD in large part on the argument that asthma leads to COPD. While it is true that some studies have shown an increased risk of developing COPD later in life if the individual had severe asthma in childhood (as high as 32 times more likely to develop compared to other types of asthmatics), there is not broad

consensus on this hypothesis and indeed there are substantial distinct differences between the two traits (i.e., asthma is generally reversible versus irreversible airflow obstruction, source of inflammation (allergies vs bacteria), response to anti-inflammatory therapeutics, etc). If the goal of the study was to identify overlap between severe childhood asthma that becomes COPD later in life, it is questionable to what extent the GWAS dataset used for this study (the UK Biobank) is ideally suited to test the hypothesis that asthma leads to COPD. That said, the authors present a reasonably comprehensive introduction (pg 2) regarding other overlapping features between asthma and COPD that justifies their overall approach, and the proof of studies of GSMB are especially compelling. Despite these concerns and questions below, overall this is a clearly written and reasonably summarized manuscript employing a novel framework for disentangling complex traits.

Major:

2.1. Further to the concern described above regarding features of overlap between asthma and COPD, did the authors consider to what extent genes identified in this study fall under categories supporting purported common features (i.e., airflow obstruction reversibility) and to what extent genes that represent distinct differences (i.e., allergic disease) did not show up?

We thank the reviewer for the suggestion. As a proof of principle, we analyzed the enrichment of asthma-specific, COPD-specific and overlap-related features in the asthma, COPD and flow central genes sets considered in this work.

We first collected allergy-related genes associated to asthma from AllerGAtlas, a recently published repository (Liu et al., Database, 2018). We then evaluated the enrichment of these gene in the asthma module and in the FC genes set (shown in the Figure below), observing that the enrichment is over three-fold higher in the asthma module with respect to the FC gene set. Additionally, the enrichment of allergy-related genes in asthma is highly significant when compared to random samples generated with DIAMOnD (as described in Sec. 4.5 of the manuscript), labeled Random(DIAMOnD) in the plot, and to the hypergeometric expected value, labeled Random(Hypergeom.). FC is instead very close to the expected hypergeometric value. Considering that FC genes will lie in the neighborhood of the asthma module, we consider this as evidence of substantial depletion of allergy-related genes in the FC set.

We repeated the same test by extracting the genes related to the emphysema phenotype from the DisGeNet repository, which is a characteristic feature of the COPD phenotype (shown below in the Figure).

As before, we find significant enrichment of emphysema genes in the COPD module and no overlap between FC genes and emphysema-related genes.

As a last test, we extracted the genes related to Airway Hyperresponsiveness (AHR) from DisGeNet, selecting the genes associated to the phenotypes "Bronchial Hyperreactivity" and "Respiratory Hypersensitivity". While AHR is a phenotype prevalently associated to asthma, there is evidence of manifestation of this phenotype in asthma-COPD overlap patients (Tkacova et al., JACI, 2016). We obtain the following overlaps:

As expected, FC and asthma genes are highly enriched in AHR-related genes. Surprisingly, we find no overlap between COPD genes and AHR genes, possibly indicating that AHR is a clinical feature that may occur in COPD patients as a result of underlying asthma-specific pathways.

2.2. GWAS database(s) - The group relied on publicly available GWAS data from the UK Biobank as the first step (compiling known GWAS loci, or sets of ‘seed’ genes, for each asthma and COPD). Can they speculate to what extent findings may have differed had they focused on GWAS data from studies specifically designed to identify genetic determinants associated with asthma and COPD? Specifically, since the most compelling data supporting the argument that asthma leads to COPD is for severe asthma during childhood, and because the authors chose to focus on an asthma candidate gene as a proof of concept which has in fact been specifically associated with childhood onset asthma (GSDMB), did they consider, at a minimum, replication of this approach in other GWAS datasets?

We would like to clarify that the genes selected as asthma and COPD disease genes (seed) are compiled from several asthma and COPD-specific GWAS studies and not from UK BioBank. For example, ORMDL3 was considered as reported in: (Torgerson et al., Nat Genet., 2011) (Li et al., J Allergy Clin Immunol., 2012), (Ferreira et al., Lancet., 2011), (Moffatt et al., N Engl J Med., 2010) (Lasky-Su et al., Clin Exp Allergy., 2012) and (Galanter et al., J Allergy Clin Immunol., 2014). All the references for the seed genes for asthma and COPD are compiled in Supplementary table S1. We use the genome-wide significant association signal from UK-Biobank (UKB) only to define the stopping criterion for the disease module construction step. We considered the UK Biobank as it has considerable statistical power because of its large sample size (500,000

individuals) to define the boundaries of the disease modules. We expect the results to be quite similar for moderate changes of the stopping criterion, since the genes with higher ranks in the DIAMOnD prioritization (i.e. with lower scores) are by design the least relevant to the initial seed gene set.

To assess the robustness of the selected boundaries, we performed the following test. We defined a range of possible small variations in the selected cutoff value iteration of DIAMOnD modules, i.e. (-30,-20,-10,-5,-1,1,5,10,20,30). For example, in the case of the asthma module the size of the module is 373 genes. When considering the variation -30 in the list, we built a perturbed asthma module by considering only the first $N - 30$ genes prioritized by DIAMOnD, obtaining a module size of $N_{\text{asthma}} - 30 = 343$ genes and do the same for COPD. We then calculated flow centrality scores between the perturbed modules, and assessed its correlation with flow centrality scores obtained on the original unperturbed modules. We repeated the same operation for all the variation values in the list (-30,-20,-10,-5,-1,1,5,10,20,30).

The results are shown below:

For all the possible variations we obtain a very high Spearman's correlation coefficient between the FCS of the perturbed modules and the FCS of the original modules (> 0.94), confirming the robustness of the FC values to moderate variations of the cutoff choice in the DIAMOnD prioritization.

We have clarified this point in the updated manuscript in Sec. "FCS stability" in Materials and Methods.

Minor:

2.3. The statement on pg 2 "people affected by asthma since birth are more likely to develop COPD at later ages" is first of all, not fully substantiated and second, needs referencing (could use references provided on pg 5 for a similar statement although better references could be selected).

We added several references to support the statement.

2.4. Figure 2a would be more informative if the actual values were placed over the bars.

We regenerated all the boxplot figures and included asterisk symbols to denote the degree of significance of the difference between the distributions: ***, ** and * respectively denote a p-value that is less than $1e-10$, $1e-4$ and 0.05.

2.5. Figure 3: what was the process for selecting the 3 examples? Was this random, or biased in some way?

The 3 examples were arbitrarily chosen among the paths with the largest number of GO annotations to visually illustrate the concept of flow central paths and show their biological relevance.

2.6. Table S1 is not readable as is.

Table S1 has been simplified and reformatted.

Other changes in the analysis:

- We found that two seed genes of asthma (LRRC32 and HLA-DQA1) were mistranslated in the symbol to entrez ID conversion step. Both the (wrong) entrez IDs were not present on the PPI, so they were excluded, while when correcting the entrez IDs we observed that one of them was present in the PPI. Thus, the seed gene total changed from 34 to 35. Nonetheless, when executing the downstream analyses we noticed negligible differences. For example, the

DIAMOND module of asthma built from the seed genes changes only by three genes out of 373, with two genes added (HLA-DQA1, CDHR3) and one removed from the module (CDH2).

- We regenerated the random samples for calculating the z-score in the flow centrality evaluation, fixing the random seed for reproducibility. The new z-scores are highly correlated to the previous one, with a spearman correlation of ~0.97.

- We regenerated and improved the clarity of most of the plots

- In the criteria for the selection of the FC paths we included an additional constraint. All the intermediate FC genes between source and destination nodes have to be traversed by a minimum of 5 shortest paths. This modification accounts for those genes whose FC scores fluctuated considerably because of their low statistics. The resulting pool of FC genes that can be selected in FC paths is reduced from 422 to 392 genes. We expect this change to make the analysis more reproducible when regenerating the random samples with different random seeds. This change is reflected in the manuscript, in the section "Flow central paths".

- In the GSDMB knockdown/overexpression section, one path has been eliminated from the results since it was no longer significant with the new criteria.

- We updated the DisGeNet annotations. We downloaded the data file from <http://www.disgenet.org/downloads> on July 19th 2019.

REVIEWERS' COMMENTS:

Reviewer #1 (Remarks to the Author):

The authors have addressed reasonably well most of the concerns raised during the last review. Few minor points that need to be addressed or require clarification.

1. The DisGeNET database that the authors use has two sets of annotations - a "known" disease-gene set (compiled from a multitude of databases) and a "BeFree" version which is based on text-mining and may contain false positives. Which version did the authors use should be made clear.
2. The authors do not quite address the comment related to the differences in the patterns of airway inflammation in asthma and COPD. For instance, how the FC approach can enable define the overlapping and transitioning phenotypes and specifically their perceived utility in the clinical diagnosis and patient management. Some discussion around these in the "Discussion" section would be useful.

Reviewer #2 (Remarks to the Author):

In this revised manuscript by Maiorino and colleagues, the authors have sought to clarify concerns raised regarding the employment of flow centrality to characterize the interactome (e.g., protein-protein interaction) common to asthma and COPD and identify mediators (e.g., disease-disease interactions) between the two diseases. A major concern with the initial version was to what extent genes identified in this study fall under categories supporting purported common features (i.e., airflow obstruction reversibility) and to what extent genes that represent distinct differences (i.e., allergic disease) did not show up. The authors are to be credited for an extensive, additional set of analyses which included leveraging existing resources (AllerGAtlas and DisGeNet to analyse the enrichment of asthma-specific, COPD-specific, and overlap features and flow central genes. It is indeed an additional strength to this manuscript (and testimony to the overall approach of this study) to discover multifold enrichment of genes in the asthma and emphysema modules, depletion of allergy-related genes in the FC set, no overlap between FC genes and emphysema-related genes, and no overlap between COPD genes and AHR genes. These additional analyses add much to this work.

A prior concern was also that the group relied on UK Biobank data to select asthma and COPD disease genes (seed), but they have clarified that in fact this information was drawn from multiple sources in the Material and Methods section and in the supplementary section. As additional added strength to this revised manuscript, they performed a test to assess the robustness of the selected boundaries and provide compelling data to defend the FC values and prioritization scheme.

There were minor concerns included substantiating the statement that "people affected by asthma since birth are more likely to develop COPD at later ages" with additional references, regeneration of boxplot figures to be more informative and Table S1. They proactively improved the quality of other figures as well.

The authors made several additional changes, including regenerating random samples for calculating the z-score, adding an additional constraint for the selection of the FC paths, and updating annotations.

We would like to thank again the reviewers for their valuable feedback. In the updated manuscript, we addressed the remaining points of the reviewers and implemented the modifications requested by the author guidelines.

Reviewers' comments:

Reviewer #1 (Remarks to the Author):

The authors of addressed reasonably well most of the concerns raised during the last review. Few minor points that need to be addressed or require clarification.

1. The DisGeNET database that the authors use has two sets of annotations - a "known" disease-gene set (compiled from a multitude of databases) and a "BeFree" version which is based on text-mining and may contain false positives. Which version did the authors use should be made clear.

We thank the reviewer for the observation. As aptly stated by the reviewer, associations coming from text mining are often characterized by high false positive rates, and for this reason the analysis was limited only to the curated gene-disease associations (i.e. BeFree GDAs were excluded). We clarified this point in the methods section.

2. The authors do not quite address the comment related to the differences in the patterns of airway inflammation in asthma and COPD. For instance, how the FC approach can enable define the overlapping and transitioning phenotypes and specifically their perceived utility in the clinical diagnosis and patient management. Some discussion around these in the "Discussion" section would be useful.

We recognize that in asthma inflammation can be either the Th2 type with eosinophils and mast cells or neutrophilic as seen with Th17 activation due to viral infection. In COPD, the inflammation is primarily neutrophilic due to cigarette smoking. A subset of smokers will get eosinophilic inflammation from smoking. While there is overlap between the inflammation in asthma and COPD, we agree with the reviewer in that they are not the same. Studying this heterogeneity in immune response demands detailed information on cell type composition in the lung tissues. Nonetheless, we believe that flow centrality is a first step to identify the potential genes involved in the overlapping pathways associated to asthma and COPD conditions, as well as transitioning or hybrid phenotypes. By definition, flow central genes are

characterized by an unexpectedly large number of interactions (direct or indirect) with asthma and COPD genes, and thus they may have a role in causing (or protecting from) the development of mixed phenotypes. Multi-omics measurements (such as transcriptomics, genomics and epigenomics) could be leveraged to define a molecular phenotype of the flow central genes in affected patients. By correlating these molecular profiles with clinical conditions and outcomes, it would be in principle possible to locate these profiles on the asthma-COPD spectrum, creating new opportunities for targeted therapeutics. One major hurdle in this direction is caused by the scarce availability of data. Asthma typically occurs since young age, while COPD develops at older ages. Thus, following the transition from a pure asthmatic pathology to a mixed asthma-COPD condition is generally unfeasible in controlled settings. Furthermore, despite the recent efforts in delineating the hallmark features of asthma and COPD (i.e. the GINA and GOLD standards), a rigorous classification of mixed phenotypes and their unique characteristics is still lacking, making the integration of data from different studies even more problematic.

We have added a part in the discussion addressing this topic.

Reviewer #2:

In this revised manuscript by Maiorino and colleagues, the authors have sought to clarify concerns raised regarding the employment of flow centrality to characterize the interactome (e.g., protein-protein interaction) common to asthma and COPD and identify mediators (e.g., disease-disease interactions) between the two diseases. A major concern with the initial version was to what extent genes identified in this study fall under categories supporting purported common features (i.e., airflow obstruction reversibility) and to what extent genes that represent distinct differences (i.e., allergic disease) did not show up. The authors are to be credited for an extensive, additional set of analyses which included leveraging existing resources (AllerGAtlas and DisGeNet to analyse the enrichment of asthma-specific, COPD-specific, and overlap features and flow central genes. It is indeed an additional strength to this manuscript (and testimony to the overall approach of this study) to discover multifold enrichment of genes in the asthma and emphysema modules, depletion of allergy-related genes in the FC set, no overlap between FC genes and emphysema-related

genes, and no overlap between COPD genes and AHR genes. These additional analyses add much to this work.

A prior concern was also that the group relied on UK Biobank data to select asthma and COPD disease genes (seed), but they have clarified that in fact this information was drawn from multiple sources in the Material and Methods section and in the supplementary section. As additional added strength to this revised manuscript, they performed a test to assess the robustness of the selected boundaries and provide compelling data to defend the FC values and prioritization scheme.

There were minor concerns included substantiating the statement that “people affected by asthma since birth are more likely to develop COPD at later ages” with additional references, regeneration of boxplot figures to be more informative and Table S1. They proactively improved the quality of other figures as well.

The authors made several additional changes, including regenerating random samples for calculating the z-score, adding an additional constraint for the selection of the FC paths, and updating annotations.

We thank the reviewer for the kind comments and valuable feedback.